# A single pair of neurons links sleep to memory consolidation in *Drosophila melanogaster*

Paula R Haynes[†], Bethany L Christmann[†], Leslie C Griffith*

Department of Biology, Volen Center for Complex Systems, National Center for Behavioral Genomics, Brandeis University, Waltham, United States

**Abstract** Sleep promotes memory consolidation in humans and many other species, but the physiological and anatomical relationships between sleep and memory remain unclear. Here, we show the dorsal paired medial (DPM) neurons, which are required for memory consolidation in *Drosophila*, are sleep-promoting inhibitory neurons. DPMs increase sleep via release of GABA onto wake-promoting mushroom body (MB) α'/β' neurons. Functional imaging demonstrates that DPM activation evokes robust increases in chloride in MB neurons, but is unable to cause detectable increases in calcium or cAMP. Downregulation of α'/β' $GABA_A$ and $GABA_B$R3 receptors results in sleep loss, suggesting these receptors are the sleep-relevant targets of DPM-mediated inhibition. Regulation of sleep by neurons necessary for consolidation suggests that these brain processes may be functionally interrelated via their shared anatomy. These findings have important implications for the mechanistic relationship between sleep and memory consolidation, arguing for a significant role of inhibitory neurotransmission in regulating these processes.

*For correspondence: griffith@brandeis.edu

†These authors contributed equally to this work

## Introduction

Accumulating evidence suggests that sleep plays a role in promoting the consolidation of memory (*Stickgold, 2005*; *Diekelmann and Born, 2010*; *Mednick et al., 2011*; *Abel et al., 2013*; *Rasch and Born, 2013*). Sleep deprivation following an associative learning task impairs consolidated memory in *Drosophila*, rodents, and humans whereas sleep immediately after a learning task actually improves consolidated memory across the same broad range of organisms (*Ganguly-Fitzgerald et al., 2006*; *Donlea et al., 2011*; *Rasch and Born, 2013*; *Diekelmann, 2014*). It is not, however, clear exactly how sleep promotes memory consolidation: it may simply be a permissive state generated by other brain regions that prevents sensory interference with memory circuits, or alternatively the memory circuitry itself may actively participate in sleep promotion as an integral aspect of the consolidation process. To begin to probe these issues, we have investigated the role of the dorsal paired medial (DPM) neurons, which are critical to memory consolidation in *Drosophila melanogaster*, in the regulation of sleep.

The *Drosophila* learning and memory circuitry has been well characterized and provides an excellent system in which to study cellular interactions between sleep and memory consolidation. The mushroom bodies (MBs) are a set of ca. 5000 neurons in the *Drosophila* brain, organized into five distinct lobular neuropils, which are required for odor memory acquisition, consolidation, and retrieval. Although the anatomy involved in memory consolidation in mammals is highly complex and distributed, in the fly it is quite compact: the DPM neurons, a single pair of neurons innervating all of the MB lobes, are the mediators of consolidation for odor memories (*Waddell et al., 2000*; *Keene et al., 2004, 2006*; *Yu et al., 2005*; *Krashes and Waddell, 2008*). Like mammals, *Drosophila* consolidates memories at the systems level. Critical memory information is transferred from short-term storage in neurons required for initial acquisition to anatomically and physiologically distinct long-term storage

**eLife digest** Sleep affects memory: if you do not sleep well after a learning task, chances are you will not be able to recall whatever you tried to learn earlier. This is seen in almost all animals ranging from the fruit fly *Drosophila,* to mice and humans. However, the precise details of how memory and sleep are connected remain unclear.

*Drosophila* is an excellent model for teasing out the connections between memory and sleep. This is because its brain has a simple and well-studied memory region that contains a pair of nerve cells called the dorsal paired medial neurons. These neurons enable memories to be stored for the long term. Here, Haynes et al. asked whether these neurons can also affect sleep, and if so, how.

The experiments show that the dorsal paired medial neurons promote sleep in fruit flies. The neurons release a signaling molecule called GABA, which is detected by a type of neighboring 'mushroom body' neuron that usually promotes wakefulness. This leads to increases in the levels of chloride ions in the mushroom body neurons, but no change in the levels of calcium ions and a molecule called cAMP, which indicates that GABA inhibits these cells. Flies that have lower levels of two receptor proteins that detect GABA sleep less than normal flies.

Haynes et al.'s findings suggest that dorsal paired medial neurons deactivate their neighbors to promote sleep in fruit flies. This result was unexpected because current models of memory formation propose that dorsal paired medial neurons can activate the mushroom body neurons. Understanding how inhibiting mushroom body neurons influences memory will require researchers to reassess these models.

sites (*Yu et al., 2005*; *Krashes et al., 2007*; *Wang et al., 2008*; *Cervantes-Sandoval et al., 2013*; *Dubnau and Chiang, 2013*). DPM neurons, along with the α′/β′ subset of MB neurons, are required for early phases of this memory information transfer (*Keene et al., 2004*, *2006*; *Krashes et al., 2007*; *Krashes and Waddell, 2008*).

The MB memory circuit has also been implicated in the regulation of sleep by a number of studies (*Joiner et al., 2006*; *Pitman et al., 2006*; *Yuan et al., 2006*; *Yi et al., 2013*). Loss of MB 5HT$_{1A}$ receptors (*Yuan et al., 2006*) as well as alterations in MB PKA activity (*Joiner et al., 2006*) and neurotransmitter release (*Pitman et al., 2006*) have been shown to affect sleep in *Drosophila* in a lobe-specific manner. Mutation of the *amnesiac (amn)* gene, which encodes a putative neuropeptide expressed in DPM neurons (*Waddell et al., 2000*), results in fragmented sleep and impaired sleep rebound following deprivation, suggesting a role for these cells (*Liu et al., 2008*). While the molecular and cellular requirements for sleep and memory clearly overlap, whether the circuit that regulates sleep is identical to that required for memory is not clear and this is a question that bears directly on the functional interrelationship between sleep and memory consolidation.

The primary question addressed in this study is the role of the DPM neurons and their outputs in regulation of sleep. The DPM contribution to memory consolidation had been suggested to occur due to the release of acetylcholine (ACh) (*Keene et al., 2004*) and the product of the *amn* gene (*Waddell et al., 2000*) enhancing MB potentiation via an excitatory feedback loop (*Yu et al., 2005*; *Keene and Waddell, 2007*) similar to what has been proposed to occur in the mammalian hippocampus (*Hebb, 1949*; *Hopfield, 1982*; *Amit, 1989*; *Treves and Rolls, 1994*; *Battaglia and Treves, 1998*; *Lisman, 1999*). Recently, however, DPM release of serotonin (5HT) has been shown to promote anesthesia resistant memory, a form of consolidated memory, by acting on Gα$_i$-coupled 5HT$_{1A}$ receptors in the α/β lobes of the MBs (*Lee et al., 2011*). The involvement of a potentially inhibitory receptor, 5HT$_{1A}$, in consolidation suggests that a simple positive feedback model for consolidation is unlikely to be completely correct, and it highlights the fact that there is currently no information on the functional nature of the synapses between DPM neurons and MBs. An understanding of this synapse is critical for elucidating DPM's role in sleep. To address this aspect of DPM function, we have investigated the nature of their connection to the MBs.

Here, we show that the DPM neurons promote sleep via the release of 5HT and the inhibitory neurotransmitter GABA. We find that DPM activation results in inhibitory chloride influx into post-synaptic MB neurons and find no evidence that DPM neuron activation has an excitatory effect on post-synaptic MB neurons. We suggest a model in which post-synaptic MB α′/β′ neurons are wake-promoting, and

inhibition by DPM neuron GABA and 5HT release during memory consolidation results in increased sleep. These findings provide new insight into the functional relationship between sleep and memory consolidation, and suggest an important role for inhibitory neurotransmission in regulating these processes.

## Results

### DPM activity promotes sleep

In order to determine whether DPMs play a role in regulating sleep, we acutely activated these neurons by driving the warmth-sensitive cation channel, dTrpA1 (*Hamada et al., 2008*) with *NP2721-GAL4*, a driver with relatively specific and strong DPM expression (*Figure 1—figure supplement 1*). A temperature shift at ZT0 from 22°C, a temperature at which dTrpA1 is inactive, to 31°C, where it is open and can depolarize DPMs, produced an immediate and dramatic increase in sleep (*Figure 1A*). dTrpA1 activation with a weaker, but even more specific DPM driver line, *VT64246-GAL4*, also resulted in immediate and significant increases in sleep (*Figure 1—figure supplement 2A,C*) indicating the effect is due to activation of DPM neurons and not other neurons in the *NP2721-GAL4* expression pattern. Activating DPMs with dTrpA1 did not alter the level of locomotor activity during waking periods (light period of first day: $P_{GAL4} = 0.57$, $P_{UAS} < 0.0001$ and for the dark period of first day: $P_{GAL4} = 0.5$, $P_{UAS} = 0.13$), suggesting that this treatment does not cause locomotor impairment. Additionally, video recordings at 0-2 min and >2 hr after DPM activation at 31–32°C show that flies are immediately arousable by gentle tapping and have normal geotaxis and locomotion, consistent with DPM activation inducing a sleep state rather than paralysis or locomotor dysfunction (*Video 1*).

Upon cessation of dTrpA1 activation, after 2 days of activation at 31°C, flies showed decreased sleep. The negative sleep rebound after release is consistent with the presence of strong compensatory homeostatic mechanisms counteracting excessive sleep (*Shang et al., 2013*) and/or excessive DPM activity. Since DPM neurons are normally activated in the first 0–3 hr following training (*Yu et al., 2005*; *Cervantes-Sandoval and Davis, 2012*), it is unlikely that these neurons would ever naturally exhibit such high levels of activity for the length of time we have imposed artificially. The immediate increase in sleep upon dTrpA1 activation is most likely to be indicative of normal DPM function. The idea that DPMs can act acutely is in agreement with the literature that shows there is a temporally circumscribed window during which they are required for consolidation (*Keene et al., 2004*, *2006*; *Yu et al., 2005*) and with the finding that a short period (45 min) of TrpM8 activation of DPM neurons following a learning task rescues age-induced memory impairment (*Tonoki and Davis, 2012*).

Since activation of DPM neurons acutely induces sleep in flies, we wanted to determine if DPM activity also played a role in the maintenance of baseline sleep. In order to assess this, we used the *NP2721-GAL4* line to drive expression of a temperature-sensitive, dominant negative Dynamin, Shibire<sup>ts</sup> (Shi<sup>ts</sup>) to block vesicle recycling in DPMs. Following a shift at ZT0 from the permissive temperature of 18°C, to the restrictive temperature of 31°C, flies showed a small but significant decrease in levels of nighttime sleep relative to the baseline sleep of each genotype at 18°C (*Figure 1B1*). Because the temperature shift protocol induced changes in the baseline nighttime sleep of control flies (compare 'baseline' and 'recovery' days in panel B1) we asked if the ability of DPM inhibition to decrease nighttime sleep was independent of baseline by doing a second round of temperature shift (*Figure 1B2*). We found that DPM inhibition decreased sleep regardless of the starting baseline. We obtained a similar result using another DPM line, *C316-GAL4* to drive Shi<sup>ts</sup> (*Figure 1—figure supplement 3*). Additionally, when the weaker, but cleaner, *VT64246-GAL4* driver line was used with the temperature-sensitive repressor, *Tubulin-GAL80<sup>ts</sup>*, to produce acute expression of the hyperpolarizing potassium channel Kir2.1 in DPM neurons nighttime sleep was also reduced (*Figure 1—figure supplement 2B,D*).

These data demonstrate that sleep loss after inhibition of DPM activity is both cell-specific and independent of the particular method used to suppress DPM activity. The limitation of the DPM loss-of-function phenotype to nighttime sleep implies there is a baseline function of DPM activity that occurs even in isolated animals in a relatively stimulus-poor environment, but the small magnitude of these changes suggests that DPMs are not the major driver of baseline sleep. The DPM-dependent gain-of-function experiments with dTrpA1, however, indicate that significant changes in both nighttime and daytime sleep can be produced with acute activation of DPM neurons, as might perhaps naturally occur secondary to some type of experience.

The DPM neurons are coupled via gap junctions to second pair of neurons innervating the MB, the anterior paired lateral (APL) neurons (*Wu et al., 2011*). It was possible that sleep gains resulting

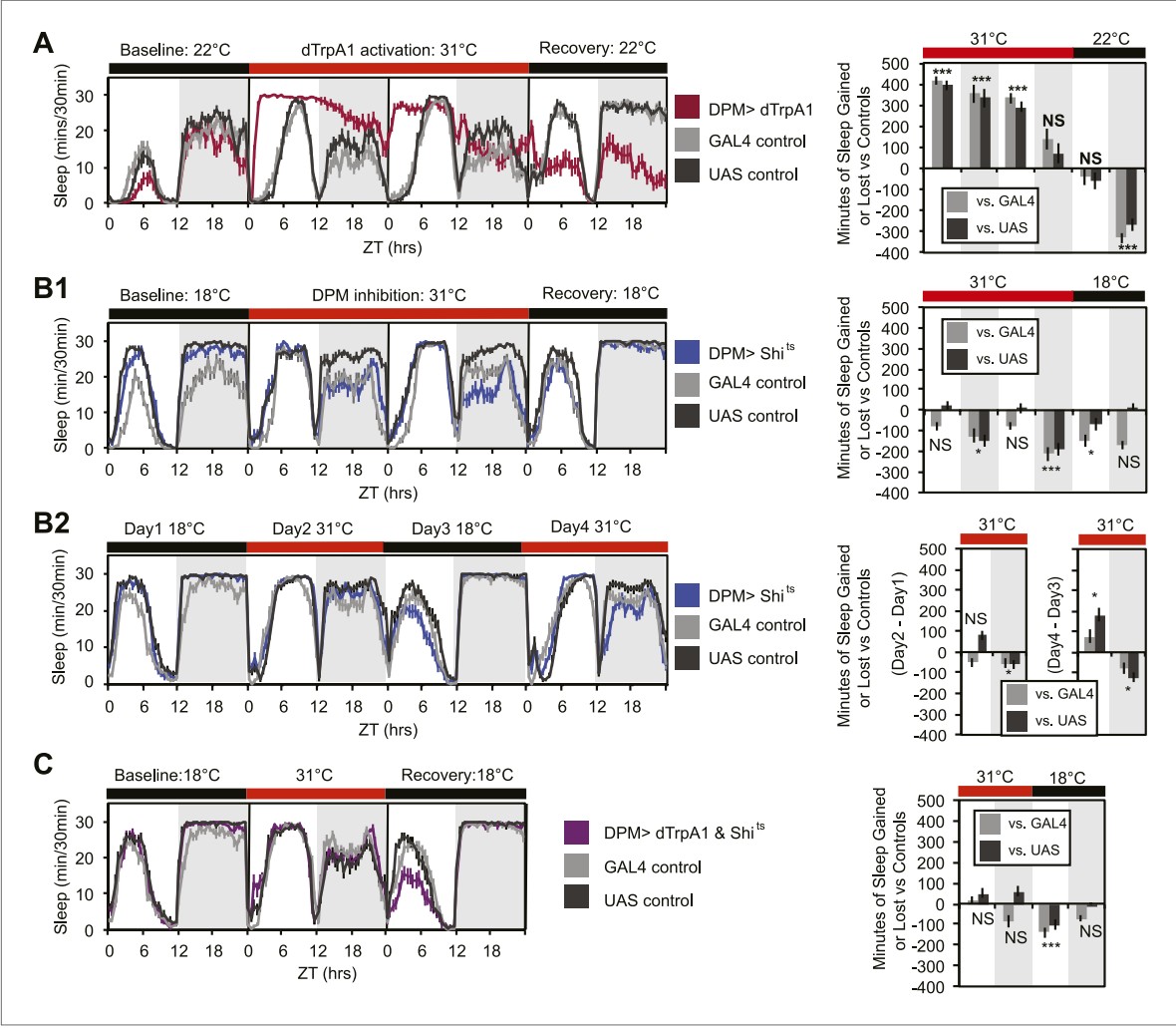

**Figure 1**. DPM activity and synaptic release are sleep promoting in a cell-autonomous manner. (**A**) Flies exhibit large gains in sleep when DPM neurons are activated with dTrpA1 at 31°C (*w-; NP2721-GAL4/ UAS-dTrpA1-II*). Compensatory sleep loss is apparent during recovery following 2 days of dTrpA1 activation. (**B1**) Flies show small but significant sleep loss when DPM synaptic release is inhibited with Shi[ts] after shift to 31°C (*w-; NP2721-GAL4; 20xUAS-IVS-Syn21-Shi[ts]*). Continuing sleep loss is apparent during the first 12 hr of recovery following Shi[ts] inhibition of DPM synaptic release. (**B2**) Sleep loss can be seen over multiple cycles of temperature shift when DPM synaptic release is inhibited with Shi[ts] (*w-; NP2721-GAL4; 20xUAS-IVS-Syn21-Shi[ts]*). For quantification in **B2**, day 1 was used as a baseline to calculate day 2 sleep changes and day 3 was used as a baseline to calculate day 4 sleep changes. (**C**) Sleep gains resulting from dTrpA1 activation are fully blocked when DPM synaptic release is inhibited with Shi[ts] at 31°C (*w-; NP2721-GAL4/ UAS-dTrpA1-II; 20xUAS-Tts-Shi[ts]*). Left plots show sleep in 30-min bins during a baseline day (22°C for dTrpA1 alone, 18°C for Shi[ts] or combined UAS experiments), followed by 1–2 days of DPM hyperactivation or inhibition (31°C) and 1 day of recovery (22°C for dTrpA1 alone, 18°C for Shi[ts] or combined UAS experiments). Right plots show a quantification of the 30-min data in 12-hr bins across 1 or 2 days of heating and 1 day of recovery. Sleep change is quantified as the minutes of sleep gained or lost by the experimental genotype in comparison to either the *UAS* or *GAL4* control genotypes during heating and recovery periods. Grey shading indicates the dark period/night, red bars indicate increased temperature. All data are presented as mean ± SEM where * represents p < 0.05, **p < 0.001, and ***p < 0.0001 using the Mann-Whitney-Wilcoxon rank sum test. Calculation of sleep gain or loss and statistics are described in the 'Materials and methods' section.

The following figure supplements are available for figure 1:

**Figure supplement 1**. Comparison of DPM-expressing *GAL4* lines used in experiments.

**Figure supplement 2**. DPM activity regulated by a different *GAL4* insertion is also sleep-promoting.

**Figure supplement 3**. Vesicle release from DPMs promotes consolidated nighttime sleep.

**Figure supplement 4**. An additional UAS transgene does not prevent dTrpA1-induced sleep gains.

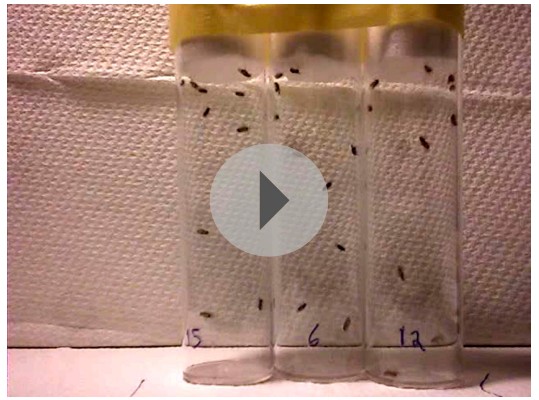

**Video 1**. DPM > dTrpA1 activation induces sleep, but not locomotor impairment. Groups of ten individual female flies with DPM*(NP2721)> dTrpA1* (center), *UAS-dTrpA1* (left) or DPM *(NP2721)-GAL4* (right) were kept at 31°C for 2 hr before video recording. Flies with DPM driven dTrpA1 expression show normal locomotion when gently tapped (0:00:01), but quickly assume a stationary resting position after ~30 s undisturbed (0:00:35), whereas control flies remain awake and continue to explore the environment. All flies were anesthetized with $CO_2$, counted, and sorted into groups of ten 1 day prior to video recording and kept on food at 22°C prior to heating. Flies were heated at 31°C on food for 2 hr and flipped to empty vials just prior to recording.

from dTrpA1-mediated activation of DPMs were not due to neurotransmitter release from DPMs themselves, but instead were a secondary result of gap–junction coupled APL activation and neurotransmitter release. In order to distinguish between these possibilities, we coexpressed dTrpA1 and the temperature-sensitive Dynamin mutant, Shi[ts], in DPM neurons. Since Shi[ts] protein and mRNA are unlikely to pass through gap junctions, DPM-Shi[ts] expression should prevent neurotransmitter release in a cell-autonomous manner from DPM, but not affect APL neurons. Thus, if dTrpA1-mediated sleep gains are the result of DPM, but not APL neurotransmitter release, they should be blocked by the coexpression of Shi[ts] in DPMs at high temperature. Conversely, if APL neurotransmitter release is responsible for sleep gains resulting from DPM activation, DPM expression of Shi[ts] should have no effect on dTrpA1-evoked sleep. We found that coexpression of Shi[ts] completely blocked activity-induced sleep gains (*Figure 1C*). This was not due to dilution of GAL4-mediated expression since the coexpression of a neutral second UAS transgene (UAS-GCaMP6) did not block dTrpA1-stimulated sleep (*Figure 1—figure supplement 4*). This suggests that sleep from dTrpA1-mediated DPM activation is the result of release of neurotransmitter from DPMs, not APLs.

Thus, we find that DPMs are capable of acutely promoting sleep and have an additional role in mediating baseline sleep during the night in stimulus-poor conditions (single flies in sleep tubes). How this function of DPMs is regulated is unknown. Given their role in memory consolidation, however, it is likely DPMs are chiefly active in acute sleep regulation when they are recruited to promote sleep following stimulus-rich experiences such as learning.

## α'/β' activity promotes wakefulness

MB α'/β' neurons are thought to be a key postsynaptic target of DPM neurons (*Keene et al., 2006*; *Krashes et al., 2007*; *Pitman et al., 2011*). Both DPM and α'/β' activity are required during the memory consolidation period 0–3 hr after training for the storage of subsequent 24 hr long-term memory (*Krashes and Waddell, 2008*). Recently, multiple groups have posited that *Drosophila* experiences a form of systems consolidation, similar to that of mammals, in which memories are transferred from a set of neurons serving as a short-term storage site (e.g., the hippocampus in mammals, and γ and α'/β' lobes in flies) to a different set of neurons which store the memory in a more stable long-term state (e.g., the cortex in mammals, α/β lobes and MB output neurons in flies) (*Cervantes-Sandoval et al., 2013*; *Dubnau and Chiang, 2013*). Since systems consolidation in *Drosophila* requires DPM and α'/β' activity and is known to be promoted by sleep in other organisms, we reasoned that α'/β' activity may also play a role in promoting sleep. While it has been shown previously that the *Drosophila* MB can promote sleep (*Joiner et al., 2006*; *Pitman et al., 2006*; *Yi et al., 2013*), a specific role for the α'/β' lobes has not been reported.

To address this issue, we acutely activated these neurons with an MB-restricted version of the α'/β' driver line *c305a-GAL4* and the warmth-sensitive cation channel, dTrpA1. If DPM neurons act to excite α'/β', as postulated by models of consolidation, we would expect this manipulation to increase sleep. Instead, we see a strong decrease in nighttime sleep. This α'/β'–dependent nighttime sleep loss remained stable throughout 48 hr of dTrpA1 activation and was accompanied by increasing daytime sleep loss which continued even after release from dTrpA1 activation (*Figure 2A,C*). This unusual pattern exactly matches the phenotype seen in flies expressing Shi[ts] in DPM neurons. Thus, DPM and α'/β'

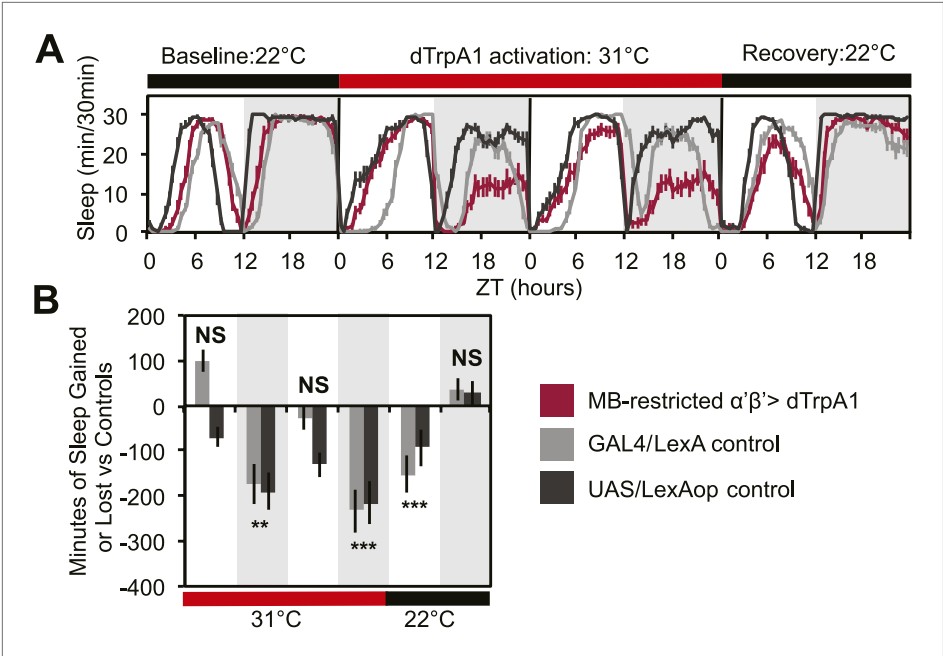

**Figure 2**. MB α'β' neuron activity promotes wakefulness. The α'β' *c305a-GAL4* driver line was crossed to *UAS-dTrpA1-II* (with *c305a* expression restricted to the MB) to determine effects on sleep of α'β' activation. (**A**) shows sleep in 30-min bins during a baseline day (22°C), followed by 2 days of DPM hyperactivation (31°C) and 1 day of recovery (22°C). (**B**) shows minutes of sleep gained or lost by the experimental genotype in comparison to either the *UAS* or *GAL4* control genotypes during heating and recovery periods. MB-restricted genotypes are: 1). *UAS-dTrpA1, ptub>GAL80>/c305a-GAL4; MB-LexA/LexAop-Flp*, 2). *c305a-GAL4; MB-LexA* (GAL4/LexA control), and 3). *UAS-dTrpA1, ptub>GAL80>; LexAop-Flp* (UAS/LexAOP control). Grey shading indicates the dark period/night, red bars indicate increased temperature. All data are presented as mean ± SEM where * represents p < 0.05, **p < 0.001 and ***p < 0.0001 using the Mann-Whitney-Wilcoxon rank sum test. Calculation of sleep gain or loss and statistics are described in the 'Materials and methods' section.

activity have opposing roles in the regulation of sleep: DPM activity promotes sleep whereas α'/β' activity is wake-promoting. These data suggest that DPM neurons may inhibit MB neurons.

## DPMs contain 5HT and GABA, but not ACh or dopamine

The shared requirement for activity during memory consolidation (*Krashes et al., 2007*; *Krashes and Waddell, 2008*) as well as the extensive physical connectivity as determined by membrane-localized GRASP (*Pitman et al., 2011*) strongly suggests that DPM and MB neurons are synaptically connected. Given the lack of information on the functional nature of the connections, we set out as a first step to determine what neurotransmitters are present in DPM neurons. Colocalization of mCD8-GFP expression in DPM cell bodies with staining against a panel of neurotransmitters shows that DPM neurons contain both GABA (*Figure 3A*) and 5HT (*Figure 3—figure supplement 1A*). DPM cell bodies also stain positively for Gad1 (*Figure 3C*), the GABA synthetic enzyme. We found no evidence for expression of ChAT, the ACh synthetic enzyme (*Figure 3C*), or tyrosine hydroxylase (TH), the rate-limiting enzyme for catecholamine synthesis (*Figure 3—figure supplement 1B*). These combined results suggest that the DPM neurons release GABA and 5HT, but not ACh or dopamine.

Although GABA release is known to be inhibitory, there are both stimulatory and inhibitory 5HT receptors in the *Drosophila* brain and it is unknown whether $G\alpha_s$-coupled 5HT receptors, such as 5HT7, are expressed in the MBs. To test whether 5HT could be stimulatory in the MBs, we used *5HT7-GAL4* (*Becnel et al., 2011*) to drive expression of Epac1-camps (EPAC) (*Nikolaev et al., 2004*; *Shafer et al., 2008*), a FRET-based cyclic nucleotide sensor. There was no expression evident in the MBs, although there was strong fluorescence in the central complex as reported previously (*Becnel et al., 2011*). We bath-applied 5HT and saw increased cAMP in the labeled cells of the ellipsoid body, confirming the efficacy of the drug as well as the positive coupling to cyclase in 5HT7-GAL4+ cells

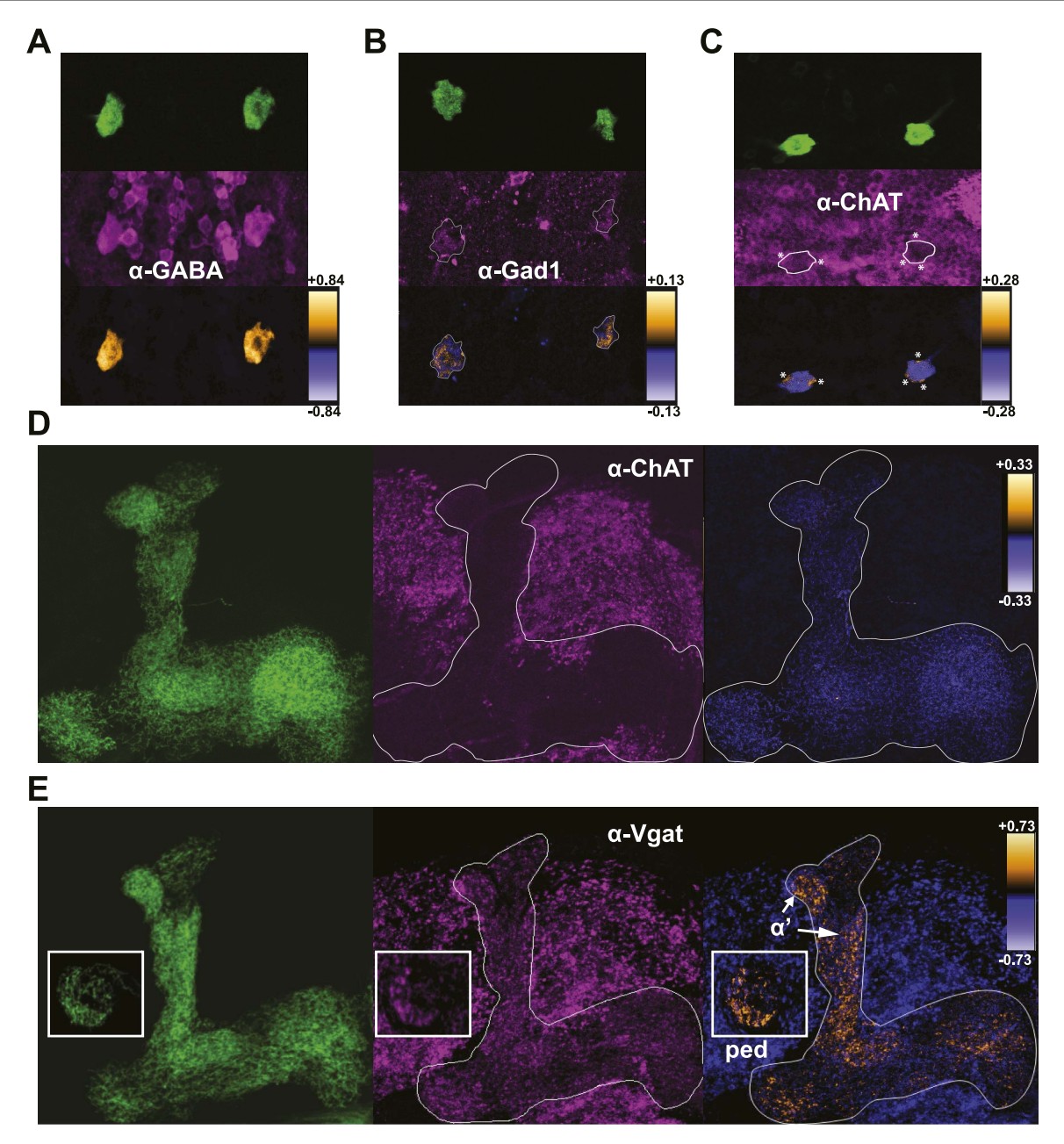

**Figure 3**. DPMs are GABAergic, but not cholinergic. (**A–C**) Top: *VT64246-GAL4* was used to drive expression of membrane-localized mCD8-GFP in DPM cell bodies, which was visualized with an anti-GFP antibody. Middle: brains were stained using antibodies against (**A**) GABA (N = 11/11 cell bodies with positive staining), (**B**) Gad1 (N = 12/13 cell bodies with positive staining), and (**C**) ChAT (N = 11/11 cell bodies had no staining). Although a number of neighboring ChAT-positive cell bodies cross over the periphery of the DPM cell bodies resulting in very localized correlation between channels (* in image), the DPMs do not show a general colocalization with anti-ChAT. Bottom: ICA was used to visualize the relative colocalization between DPM>GFP and transmitter staining in pairs of DPMs. (**D–E**) Left: *VT64246-GAL4* was used to drive expression of a presynaptic marker, BRP-short-GFP, in DPM projections to the MB. Middle: brains were stained with antibodies against (**D**) ChAT (N = 16/16 MB lobe sets with negative staining) and (**E**) VGAT, with insets showing the MB peduncles (N = 10/10 MB lobe sets with positive staining). Right: ICA was used to build false color maps of relative colocalization between DPM>Brps-GFP and transmitter staining in DPM projections. For ICA, orange indicates colocalization/correlation of pixel intensities between channels (PDM>0) and purple indicates a lack of colocalization/anticorrelation of pixel intensities between channels (PDM<0) relative to the scale shown for each image (see 'Materials and methods' for further details). 'α'' indicates the MB α' lobe and 'ped' indicates peduncles, shown in the inset.

*Figure 3. Continued on next page*

Figure 3. Continued

The following figure supplements are available for figure 3:

**Figure supplement 1**. DPMs are serotonergic, but not dopaminergic.

**Figure supplement 2**. MB neurons do not express stimulatory serotonin receptors.

(*Figure 3—figure supplement 2*). Because the lack of *5HT7-GAL4* expression in MB does not necessarily mean there is not a stimulatory 5HT receptor expressed there, we also bath-applied 5HT to brains with EPAC driven by *MB247-lexA* to determine if there was an excitatory response from this structure. We saw no increase in cAMP (*Figure 3—figure supplement, 2*). Because EPAC may not be not effective at reporting inhibition (e.g., if there is no basal activation of cyclase), we cannot rule out inhibitory effects of 5HT via $5HT_{1A}$, which is known to be expressed in MBs (*Yuan et al., 2006*; *Lee et al., 2011*).

DPMs project throughout the MBs and have both pre- and postsynaptic markers comingled in all lobes (*Waddell et al., 2000*; *Wu et al., 2013*) To investigate where DPM GABA might be released, we examined colocalization of a DPM-expressed presynaptic marker, Bruchpilot-short-GFP (*Schmid et al., 2008*; *Fouquet et al., 2009*) with immunostaining against VGAT, the vesicular GABA transporter. Colocalization was prominent in the MB α′/β′ lobes and MB peduncles (*Figure 3E*). This is consistent with a role of DPM neurons in inhibiting α′/β′ neurons in order to promote sleep and opens up the interesting possibility that there may be branch-specific neurotransmission from DPM neurons (*Yu et al., 2005*; *Cervantes-Sandoval and Davis, 2012*; *Samano et al., 2012*). We found little to no colocalization between DPM presynaptic sites and the cholinergic marker ChAT (*Figure 3D*), again suggesting that DPM neurons do not release ACh.

## DPM activation has an inhibitory effect on MB neurons

These results suggest that DPM neurons might be inhibitory rather than excitatory, and are inconsistent with a role for DPM neurons in directly enhancing potentiation. To test the sign of the connection, we first used functional imaging techniques to determine if DPM activation could stimulate postsynaptic MB neurons (*Figure 4A*). We expressed the mammalian ATP-gated P2X2 receptor (*Lima and Miesenbock, 2005*; *Yao et al., 2012*) in DPMs using *NP2721-GAL4* or *c316-GAL4*, and activated these cells by applying ATP to dissected adult *Drosophila* brains. We first confirmed that bath-applied ATP was sufficient to activate the P2X2 receptors in DPMs by co-expressing genetically-encoded fluorescent sensors and using functional imaging to observe changes in fluorescence indicating a response. Using *c316-GAL4* to drive *UAS-GCaMP3.0* (*Tian et al., 2009*), *UAS-Arclight* (*Cao et al., 2013*), and *UAS-Synapto-pHlorin* (SpH) (*Meisenbock et al., 1998*), we found that P2X2-mediated stimulation effectively activated DPM neurons, evoking increases in intracellular calcium, membrane voltage, and vesicle fusion, respectively (*Figure 4B*). It should be noted that these responses were observed in the DPM projections to the MBs, not the DPM cell bodies, demonstrating that this technique successfully activates the DPMs and causes them to release neurotransmitter from their projections onto downstream targets in the MB neuropil. We also co-expressed P2X2 receptors and *UAS-GCaMP3.0* in the DPMs using *NP2721-GAL4* to confirm that this technique was effective with the weaker driver (*Figure 4—figure supplement 1A*).

To determine the effect of DPM activation on the MBs, we expressed P2X2 receptors in DPMs using the *UAS/GAL4* binary expression system and either GCaMP3.0 or EPAC in the MBs using the *lexA/lexAop* binary expression system to observe changes in intracellular calcium or cAMP, respectively (*Figure 4A*). We found that activation of DPM neurons had no excitatory effects on the MBs when using either *NP2721-GAL4* (*Figure 4—figure supplement 1B,C*) or the stronger *c316-GAL4* with *eyeless-GAL80* and *MB-GAL80* to restrict expression to DPM neurons (*Figure 4C,D*). To confirm that these negative results were not due to lack of drug efficacy or some systematic problem, positive controls (P2X2 and GCaMP3.0 or EPAC expressed in the same cell type) were performed concurrently with every experiment (data not shown). We also did a separate set of experiments using dTrpA1 and a temperature step to activate DPM neurons, but failed to see any MB calcium responses (data not shown).

Although we demonstrated that P2X2 is capable of activating DPM neurons and causing vesicle fusion, we needed to rule out the possibility that MBs were simply incapable of responding to excitatory

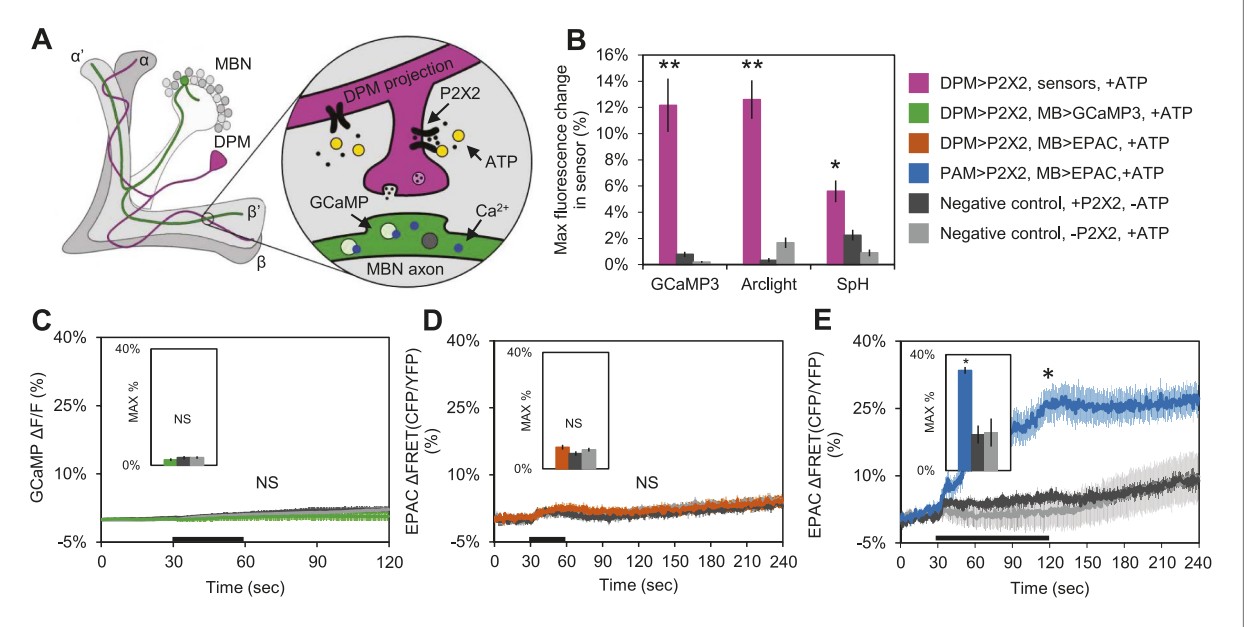

**Figure 4.** DPM activation has no excitatory effect on the MBs. (**A**) Schematic of genetic set-up for functional imaging experiments. P2X2 receptors were expressed in the DPM neurons, and fluorescent sensors (GCaMP shown) were expressed in the MB neurons. (**B**) Bath-applied ATP is effective at activating DPMs expressing P2X2 receptors. Mean maximum percentage change in GCaMP3.0 (*w-, eyeless-GAL80; UAS-GCaMP3.0/+; c316-GAL4/ UAS-P2X2*, N = 9 with *UAS-P2X2* transgene, 8 without [9, 8], p < 0.001 by Mann–Whitney U test), Arclight (*w-, eyeless-GAL80; UAS-Arclight/+; c316-GAL4/UAS-P2X2*, N = [11,6], p < 0.001 by Mann–Whitney U test), and Synapto-pHluorin (*w-, eyeless-GAL80; UAS-Synapto-pHluorin/+; c316-GAL4/ UAS-P2X2*, N = [11, 6], p = 0.002 for Mann–Whitney U test) fluorescence in horizontal DPM neuron projections in response to 30 s perfusions of 2.5 mM ATP. (**C–E**) Black bar denotes time of perfusion of 2.5 mM ATP or vehicle. Insets are histograms summarizing the mean maximum percentage change in fluorescence of the respective sensor. The *eyeless-GAL80* and *MB247-GAL80* transgenes were used to restrict GAL4 driven expression to DPMs. (**C**) Mean GCaMP3.0 response traces of *w-, eyeless-GAL80; lexAop-GCaMP3.0/MB247-GAL80; c316-GAL4, MB247-lexA/UAS-P2X2* (green), or without the *UAS-P2X2* transgene (grey), to 30 s perfusion of 2.5 mM ATP or vehicle (black). N = [10, 8], p > 0.05 for Kruskal–Wallis one-way ANOVA, histogram values are 1.9 ± 0.5% (green), 2.6 ± 0.5% (black), 2.7 ± 0.5% (grey). (**D**) Mean EPAC response traces of *w-, eyeless-GAL80; lexAop-EPAC/MB247-GAL80; c316-GAL4, MB247-lexA/UAS-P2X2* (orange), or without the *UAS-P2X2* transgene (grey), to 30 s perfusion of 2.5 mM ATP or vehicle (black). N = [10, 8], p > 0.05 for Kruskal–Wallis one-way ANOVA, histogram values are 7.4 ± 0.8% (orange), 5.3 ± 0.8% (black), 6.6 ± 0.7% (grey). (**E**) MBs respond to excitatory inputs. Mean EPAC response traces of *w-; R58E02-lexA/lexAop-P2X2; MB247-GAL4/UAS-EPAC* (blue), or without the *lexAop-P2X2* transgene (grey), to 90 s perfusion of 2.5 mM ATP or vehicle (black). N = [9, 5], p < 0.001 for Mann–Whitney U test, histogram values are 34.6 ± 3.1% (blue), 12.5 ± 1.2% (black), 13.2 ± 4.8% (grey). (**C–E**) Traces represent ROIs taken from horizontal sections of MB lobes. (**C–D**) ROIs were also taken from the vertical lobes and no change in fluorescence was seen (data not shown).

The following figure supplements are available for figure 4:

**Figure supplement 1**. DPM activation has no excitatory effect on the MBs.

**Figure supplement 2**. Bath-applied carbachol (CCh) evokes an excitatory response in MBs.

synaptic inputs. We first examined the response of the MBs to bath-applied carbachol (CCh), a cholinergic agonist. Tetrodotoxin (TTX) was present to block action potentials so that we could isolate direct effects on MBs. We observed robust increases in calcium and cAMP (*Figure 4—figure supplement 2*), indicating that MB neurons are capable of responding to ACh in an excitatory manner.

To show that MBs could respond to activation of upstream stimulatory neurons using the P2X2 technique, we expressed P2X2 receptors in the PAM cluster of dopaminergic neurons using *R58E02-lexA* and EPAC in the MBs using *MB247-GAL4*. Previous reports have demonstrated that the PAM cluster of dopamine neurons signals to the MBs by activation of Gα$_s$-coupled receptors (*Liu et al., 2012*). As expected, we observed an increase in cAMP in the MBs in response to PAM neuron activation (*Figure 4E*), indicating that MBs are capable of responding to stimulatory inputs.

These results clearly demonstrate that DPM neurons are not excitatory; however, these fluorescent sensors may not be effective at reporting inhibition unless there is some basal level of activity in the

circuit. Therefore, to determine if DPM activation has an inhibitory effect on MBs, we used P2X2 receptors to activate the DPMs and expressed the fluorescent intracellular chloride sensor SuperClomeleon (*Grimley et al., 2013*) in the MBs. We found that DPM activation evoked an increase in chloride in the MBs which could be almost completely blocked by bath-application of picrotoxin (*Figure 5A*). To determine if these results could be caused by DPM GABA release, we bath-applied GABA and observed similar MB SuperClomeleon responses in the presence of TTX, which could be completely blocked by picrotoxin (*Figure 5B*). These results demonstrate that DPM neurons inhibit the MBs via activation of GABA$_A$ receptors.

## GABA and 5HT mediate the sleep-promoting effects of DPMs

To determine if GABA- and/or 5HT-mediated inhibition was playing a role in the ability of DPMs to promote sleep, we manipulated transmitter in DPM neurons and receptors in α′/β′ neurons. In order to assess whether DPM GABA release promotes sleep we expressed *VGAT-RNAi* to knock down the vesicular GABA transporter, using two different DPM lines, *c316-GAL4* and *NP2721-GAL4*. Knockdown of VGAT in DPM neurons results in the loss of a large proportion of nighttime sleep (*Figure 6A,B*). Nighttime sleep loss is not due to hyperactivity since flies exhibit normal levels of nighttime activity while awake when compared to controls (for *c316-GAL4* with *VGAT-RNAi* $P_{GAL4}$ = 0.83, $P_{UAS}$ = 0.62 and for *NP2721-GAL4* with *VGAT-RNAi* $P_{GAL4}$ = 0.84, $P_{UAS}$ = 0.69).

As noted above, DPM neurons are coupled via gap junctions to the Anterior Paired Lateral (APL) neurons (*Wu et al., 2011*), which densely innervate the MB and synthesize the neurotransmitters GABA (*Liu and Davis, 2009*) and octopamine (*Wu et al., 2013*). Small molecules including neurotransmitters can pass through gap junctions (*Vaney et al., 1998*) and, for some subtypes of mammalian connexins, even RNAi fragments can pass (*Valiunas et al., 2005*). This raised the possibility that DPM sleep phenotypes could be dependent on APL-synthesized GABA and that manipulation of VGAT in DPMs might be indirectly acting by inhibition of APL GABA packaging. In order to assess this, we expressed *VGAT-RNAi* using three different APL *GAL4* lines: *GH146*, *NP5288*, *NP2631* (*Figure 6— figure supplement 1*). We never saw sleep loss with *NP2631-GAL4*. With *GH146-* and *NP5288-GAL4*, we saw weak nighttime sleep loss. When it was observed, nighttime sleep loss due to APL-driven

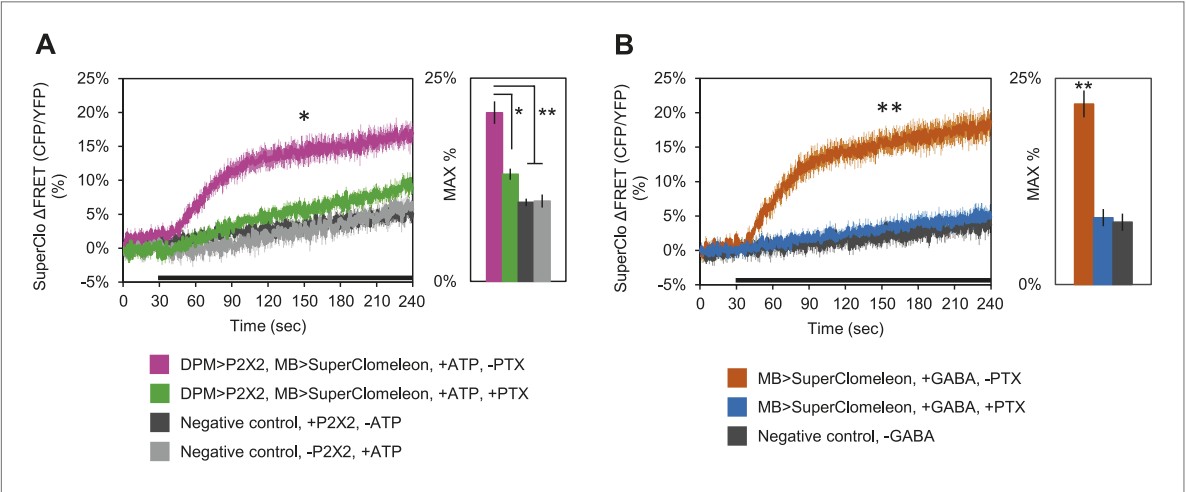

**Figure 5**. DPM activation has an inhibitory effect on the MBs. (**A**) DPM activation evokes a chloride increase in MBs via that can be reduced by picrotoxin (PTX). Mean SuperClomeleon response traces of *w-, eyeless-GAL80; lexAop-SuperClomeleon/MB247-GAL80; c316-GAL4, MB247-lexA/UAS-P2X2* to perfusion of 2.5 mM ATP alone (pink, N = 12) or in bath of 10 µM PTX (green, 10). Negative controls: mean response to ATP without *UAS-P2X2* transgene (grey, 8), or vehicle (black, 12). p < 0.001 between pink and negative controls, p = 0.001 between pink and green, p < 0.01 between green and negative controls for Mann–Whitney U test. Histogram values are 20.8 ± 1.4% (pink), 13.2 ± 0.7% (green), 9.7 ± 0.5% (black), 9.9 ± 0.8% (grey). (**B**) Bath-application of GABA in the presence of TTX evokes a chloride increase in MBs that can be blocked by PTX. Mean SuperClomeleon response traces of *w-; lexAop-SuperClomeleon/+; MB247-lexA/+* to perfusion of 1.5 mM GABA alone (orange, 8) or with 10 µM PTX (blue, 8), in 1 µM TTX bath. Negative control: Mean response to vehicle 1 µM TTX bath (black, 8). p < 0.001 for Mann–Whitney U test. Histogram values are 21.9 ± 1.6% (orange), 8.1 ± 1.1% (blue), 7.6 ± 1.0% (black). (**A**–**B**) Black bar denotes time of perfusion. Histograms summarize the mean maximum percent change in fluorescence of SuperClomeleon. Traces represent ROIs taken from vertical sections of MB lobes. ROIs were also taken from the horizontal lobes and similar results were seen (data not shown).

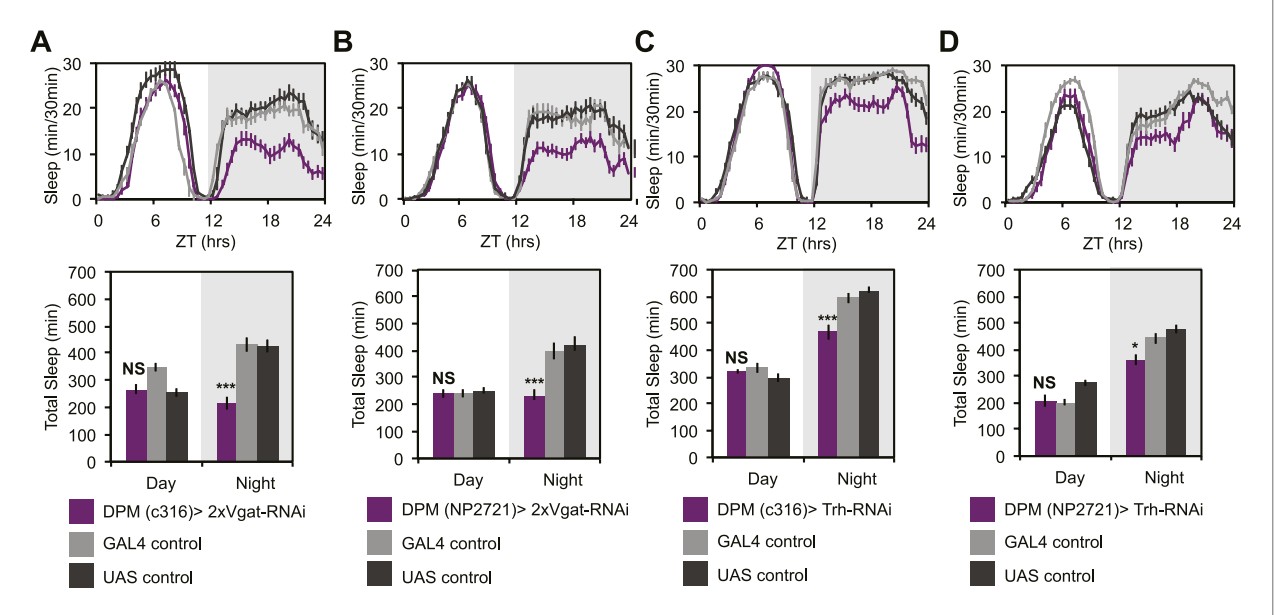

**Figure 6**. DPM GABA and 5HT promote nighttime sleep. DPM expression of VGAT was reduced by combining two copies of *UAS-VGAT-RNAi* with each of two different DPM-GAL4 drivers, *c316-GAL4* (**A**) and *NP2721-GAL4* (**B**). Expression levels of TRH where reduced in DPMs by driving *UAS-Trh-RNAi* with each of two different DPM-GAL4 drivers, *c316-GAL4* (**C**) and *NP2721-GAL4* (**D**). Top: shows total sleep in 30-min bins averaged across 3 days. Bottom: shows the same data quantified in 12-hr day/night bins. In all cases, a decrease in VGAT or 5HT synthetic enzymes (TRH) in DPMs resulted in nighttime sleep loss, with no change in nighttime activity while awake, although increases in daytime activity while awake were often apparent. Grey shading indicates the dark period/night. All data are presented as mean ± SEM where * represents $p < 0.05$, **$p < 0.001$ and ***$p < 0.0001$ using the Mann-Whitney-Wilcoxon rank sum test. Statistics are described in the 'Materials and methods' section.

The following figure supplement is available for figure 6:

**Figure supplement 1**. APL GABA can sometimes promote nighttime sleep.

expression of VGAT-RNAi was generally accompanied by an increase in activity while awake indicating an effect on locomotion; this is never seen with VGAT-RNAi in DPMs. Thus while APL GABA may be weakly sleep-promoting, its effects are qualitatively and quantitatively distinct from GABA released from DPMs. This also implies that the contribution of APLs to phenotypes seen after DPM activation or hyperpolarization is likely to be minimal. Along these lines it is also important to note that driving Shi[ts] in DPMs reduces baseline sleep and completely blocks dTrpA1-induced sleep gains. These manipulations, using DPM-expressed Shi[ts], would not be expected to influence APL activity in any way, again supporting the idea that DPMs promote sleep in a distinct and cell-autonomous manner.

While there are two known types of GABAergic neurons innervating the MB, DPMs are likely the sole source of MB lobe 5HT (*Lee et al., 2011*). MB expression of inhibitory, $G\alpha_i$-coupled $5HT_{1A}$ receptors are necessary for anesthesia-resistant memory (*Lee et al., 2011*), and are also known to promote sleep (*Yuan et al., 2006*). 5HT from DPMs, however, has not actually been shown to promote sleep. Knockdown of 5HT synthesis in DPM neurons with RNAi targeted against tryptophan hydroxylase (Trh) using the most strongly expressing DPM line, *c316-GAL4*, results in a significant loss of nighttime sleep (*Figure 6C*). This nighttime sleep loss is also apparent, but milder, with *Trh-RNAi* being driven by the somewhat weaker *NP2721-GAL4* (*Figure 6D*). No difference is seen in nighttime activity during waking periods vs controls (for *c316-GAL4* with *Trh-RNAi* $P_{GAL4} = 0.09$, $P_{UAS} = 0.17$) indicating the sleep loss is not an artifact of hyperactivity.

## DPM GABA acts on α′/β′ lobes to promote sleep

GCaMP, EPAC and Arclight experiments demonstrate a lack of excitatory transmission from DPM neurons to the MBs and SuperClomeleon experiments demonstrate that the DPMs are capable of inhibiting MB neurons. The wake-promoting phenotype of MB α′/β′ neurons as well as a shared temporal role in memory consolidation suggest these neurons could be the targets of sleep-promoting DPM

GABA release. In order to test this possibility, we expressed RNAi against *Drosophila* GABA receptors in MB α'/β' neurons. It has previously been shown that the *Drosophila* ionotropic GABA$_A$ receptor, Rdl, is highly expressed in all lobes of the MBs (*Liu et al., 2007*). Consistent with the phenotype of DPM VGAT knockdown, we observe decreased nighttime sleep with knockdown of either Rdl (*Figure 7A*), or GABA$_B$-R3 (*Figure 7B*) in MB α'/β' neurons. In both cases sleep loss is the result of a decrease in the duration of nighttime sleep episodes. Knockdown of Rdl results in less total sleep loss since an increase in the total number of nighttime sleep episodes partially compensates for the decrease in mean sleep episode duration. Importantly, concurrent expression of the *MB-GAL80* transgene, which blocks GAL4-mediated expression of receptor RNAis, greatly suppresses sleep loss and fragmentation phenotypes showing that the effects are specific to the MB α'/β' lobes (*Figure 7—figure supplement 1*). Interestingly, experiments to determine the lobe-specific role of 5HT$_{1A}$ receptors suggest that 5HT acts generally in the MB, not just on the α'/β' lobes (data not shown), suggesting that these two transmitters may play somewhat different roles at the circuit level in sleep and memory consolidation.

## Discussion

The inhibitory neurotransmitter, GABA, is known to promote sleep in both mammals (*Rasch and Born, 2013*) and *Drosophila* (*Agosto et al., 2008*), but specific sleep-promoting GABAergic neurons have not been identified in the fly. Additionally, it is known that sleep promotes memory consolidation in *Drosophila* (*Donlea et al., 2011*) and other animals (*Rasch and Born, 2013*; *Diekelmann, 2014*), but it is unclear how sleep and memory circuits interact to facilitate memory consolidation. Here we find a shared anatomical locus of memory consolidation and GABAergic sleep-promotion in the DPM neurons. We show that α'/β' neurons, postsynaptic targets of the GABAergic DPM processes, are wake-promoting. The specific involvement of neurons required for memory consolidation (and their memory-relevant post-synaptic targets) in the regulation of sleep suggests that generation of sleep by activation of learning circuits is an intrinsic property of the circuit, not an extrinsically imposed phenomenon. Further, our finding that the memory-consolidation specific DPM neurons are inhibitory suggests that inhibitory neurotransmitters may play an as-of-yet uncharacterized role in memory consolidation in *Drosophila*.

### The role of DPM vs APL neurons in regulation of sleep

Previous studies on the role of GABA in the *Drosophila* learning circuit have focused on the role of the APLs, a pair of GABAergic neurons which densely innervate the MBs and are coupled to DPMs by gap junctions. APL GABA has been shown to inhibit acquisition (*Liu and Davis, 2009*), perhaps by acting at the level of olfactory coding (*Lin et al., 2014*). APLs have also been shown to be critical for a labile component of anesthesia-sensitive intermediate-term memory but not for consolidation to long-term memory (*Pitman et al., 2011*). Because of the gap junction coupling, it was formally possible that GABA found in DPMs could be coming from APLs and that sleep loss due to DPM VGAT-RNAi expression was the result of reductions in APL VGAT levels (*Vaney et al., 1998*; *Valiunas et al., 2005*). It was also possible that phenotypes seen after manipulation of DPM electrical activity were secondary to changes in APL activity. However, a number of lines of evidence suggest that DPM neurons are intrinsically GABAergic and sleep-promoting independent of APLs. First, we find that DPMs stain positively for the GABAergic markers Gad1 and VGAT, meaning that DPMs intrinsically possess the ability to synthesize and release GABA. Second, we find that direct expression of *VGAT-RNAi* in APL neurons has a relatively minor effect on sleep as compared to phenotypes seen with expression in DPMs, indicating that GABA endogenous to DPMs is a more significant regulator of sleep than GABA from APLs. Third, we find that sleep loss due to VGAT-RNAi expression in APLs, when it is seen, is accompanied by increases in nighttime activity while awake, which is never seen with DPM-driven VGAT-RNAi expression. This indicates that although APL GABA may promote sleep in its own right, the APL VGAT-RNAi sleep loss phenotype is distinct from that of DPMs. Fourth, expression of Shi$^{ts}$ in DPMs, a manipulation that should have no effect on APL activity since it does not alter the electrical properties of DPMs, results in the same nighttime sleep loss as DPM-driven *VGAT-RNAi* and Kir2.1. Fifth, coexpression of Shi$^{ts}$ along with dTrpA1 in DPMs, a manipulation that should not affect APL neurotransmitter release, results in a complete blockade of activation-induced sleep gains. The most parsimonious explanation for all of these data is that the relevant effect of these manipulations is a change in transmitter release specifically from DPMs and that this bidirectionally modulates sleep. Thus, while it remains possible that APL neurons are modestly sleep-promoting in their

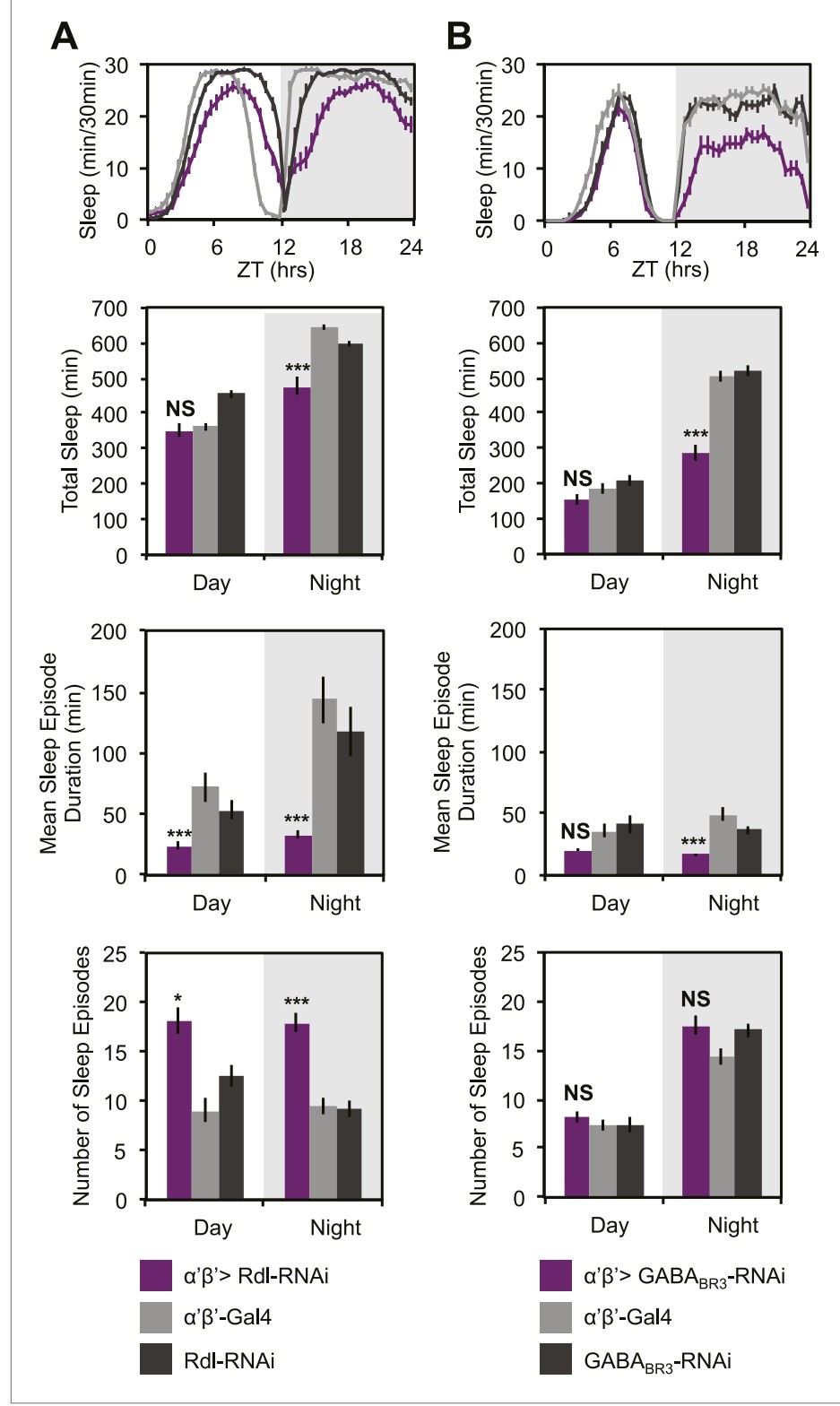

**Figure 7**. MB α'β' GABA receptors promote nighttime sleep. *c305a-GAL4* was used to drive expression of *Rdl-RNAi* (**A**) or *GABA$_{BR3}$-RNAi* (**B**) in the α'β' neurons. Top: shows total sleep in 30-min bins averaged across 3 days. Middle and bottom plots: show 3-day means of total sleep, mean sleep episode duration and number of sleep episodes quantified in 12-hr day/night bins. α'β'>*Rdl-RNAi* causes mild sleep loss and increases in nighttime sleep
*Figure 7. Continued on next page*

*Figure 7. Continued*

fragmentation, whereas α'β'>*GABA*$_{BR3}$-*RNAi* causes greater reductions in total sleep due to a decrease in the average sleep episode length. Grey shading indicates the dark period/night. All data are presented as mean ± SEM where * represents p < 0.05, **p < 0.001 and ***p < 0.0001 using the Mann-Whitney-Wilcoxon rank sum test. Statistics are described in the 'Materials and methods' section.

The following figure supplement is available for figure 7:

**Figure supplement 1**. Sleep loss resulting from *Rdl/GABA*$_{BR3}$-*RNAi* is primarily due to MB α'β' expression.

own right, we find strong evidence for an independent and significant role in regulation of sleep by DPM neurons.

## Evidence for DPM inhibition, but not excitation of α'/β' lobes

Our finding that DPM neuron activation has an inhibitory effect on post-synaptic MB neurons is in apparent disagreement with models of memory consolidation that posit recurrent excitatory feedback between DPM and α'/β' neurons (*Yu et al., 2005*; *Keene and Waddell, 2007*). Our observations using SuperClomeleon demonstrate that DPMs evoke a chloride increase in MB neurons, but we did not see decreases in calcium, membrane voltage or cAMP after DPM activation using fluorescent sensors specific for those cellular parameters. This failure is not surprising for two reasons. First, in order to see an inhibitory response it is likely that there has to be some activity or tone in the system. In cases where MB inhibition has been seen with GCaMP it is always in context of a temporally controlled acute activation of the system (*Lei et al., 2013*; *Lin et al., 2014*; *Masuda-Nakagawa et al., 2014*). Second, the nature of our DPM activation (bath application of ATP) would make it difficult to see small inhibitory changes over noise in averaged data since our activation of cells is not cleanly time-locked due to differences in diffusion of ATP into the brain between experiments. This is particularly critical with a sensor like Arclight where the expected hyperpolarization induced by inhibition might only be a few millivolts as opposed to 10–40 mV for depolarization by an action potential.

An additional critical test of the sign of this synapse is to ask if there is evidence of inhibition in the functional output of the circuit. Our finding that DPM and α'/β' activity have opposite roles in the regulation of sleep to the extent that suppression of DPMs with either Shi[ts] or Kir2.1 almost exactly phenocopies the increasing levels of nighttime sleep loss that results from dTrpA1 activation of α'/β' strongly suggests an inhibitory connection. Further, we find that decreases in DPM VGAT result in similar nighttime sleep loss phenotypes as α'/β' Rdl or GABA$_B$R3 knockdown. Thus, suppression of DPM synaptic release (Shi[ts]), DPM electrical activity (Kir2.1) and DPM GABA release (*VGAT-RNAi*) all result in nighttime sleep loss phenotypes nearly identical to dTrpA1 activation of α'/β' or loss of α'/β' GABA receptors. All of these results are consistent with a model in which DPMs act to inhibit α'/β' neurons by release of GABA.

## Inhibition of MBs and sleep

MBs are a sensory integration center in the insect brain. They have been shown, using many different behavioral paradigms, to be critical nodes for attention and arousal (*Xi et al., 2008*; *van Swinderen et al., 2009*; *Chow et al., 2011*). Paying attention to the right features of one's environment, whether naively or as a learned response, has high survival value. The fact that MBs are important sites of plasticity is likely related to this role in attention. The linkage of attention and arousal to MB output is consistent with our finding that suppression of arousal-promoting MB subsets increases sleep.

The involvement of DPMs, a neuron type previously believed to function exclusively as a regulator of memory consolidation, in control of sleep raises several interesting questions. First, are DPMs the only sleep-inducing regulators of MB activity? This seems unlikely, since memory consolidation is but one function of the MBs and there are other behavioral situations in which an animal might want to modulate MB-regulated arousal such as during courtship/aggressive behaviors (*Baier et al., 2002*; *Sakai and Kitamoto, 2006*), or more generally during the integration of internal and external cues and decision making (*Zhang et al., 2007*; *Neckameyer and Matsuo, 2008*; *Krashes et al., 2009*; *Donlea et al., 2012*; *Bracker et al., 2013*). A second question is whether DPMs might also have a role in regulating MB output in contexts other than during memory storage. Our data suggest a role for DPMs in maintaining basal levels of nighttime sleep indicating that they may be responsive to other types of

input. An understanding of the regulation of DPMs and their in vivo activity patterns will be required to gain insight into these issues.

## Inhibition in memory consolidation

A potential role for GABA-mediated inhibition in memory consolidation is novel. Long-term memory storage in mammals is believed to involve a transfer of information from one brain region to another. In *Drosophila*, it is associated with sequential potentiation of activity in specific MB neuropils. Elegant studies using conditional inhibition of transmitter release have provided a temporal ordering of transfer (*Keene and Waddell, 2007*; *Cervantes-Sandoval et al., 2013*; *Dubnau and Chiang, 2013*) leading to the idea that the role of DPM neurons is to facilitate the movement of memory from α′/β′ lobes, one initial site of memory storage, to a more permanent home in α/β lobes and perhaps other neurons (*Chen et al., 2012*). Given the clear requirements for molecular pathways associated with synaptic potentiation, and the need for synaptic transmission from both DPMs and α′/β′ neurons after acquisition, the view that information transfer involves a positive feedback loop between these cell types makes sense.

The data presented here, however, strongly suggest that the DPM neurons are inhibitory. It is clear that potentiated output from α′/β′ neurons is required for consolidation, so how could their inhibition facilitate this process? Our findings suggest DPMs are unlikely to participate directly in the excitatory arm of a recurrent feedback loop. However, both models and physiological data related to mammalian cortical/hippocampal recurrent feedback circuits involved in the maintenance of stable memory states also require the presence of inhibition (*Buzsaki and Chrobak, 1995*; *Hasselmo et al., 1995*; *Battaglia and Treves, 1998*; *Chance and Abbott, 2000*). Interestingly, the coordination of excitatory activity amongst diverse functional sub-circuits in the mammalian hippocampus and neocortex is regulated by the activity of broadly-projecting, gap-junction coupled inhibitory neurons/networks (*Freund and Antal, 1988*; *Gibson et al., 1999*; *Tamas et al., 2000*; *Baude et al., 2007*; *Jinno et al., 2007*) which have been proposed to potentially control the timing of memory replay events during sleep/memory consolidation (*Viney et al., 2013*). The DPM-APL network may represent an analogous set of neurons in *Drosophila* which function to coordinate the temporal stabilization, gating and transfer of different memory stages between different sub-circuits within the MBs. While APL neurons have been shown to broadly inhibit recurrent feedback from all MB Kenyon cells to all MB Kenyon cells (*Lin et al., 2014*), it has been proposed that DPMs may impose a directionality on internal MB feedback which would allow for memory transfer and consolidation (*Yu et al., 2005*). Although, inhibition was not previously considered for such a role, it may be capable of coordinating the timing of prime lobe output in a way that is not possible via excitation. Thus, our data are not inconsistent with the presence of an excitatory recurrent feedback loop within the MB α′/β′ lobes, but rather provide information that constrains future models in a new way and suggests new possibilities for the role of DPMs that may not have been considered previously.

What kind of role could inhibitory neurons play? While α′/β′ neurons need to be active during acquisition, DPMs do not and this temporal difference suggests some testable possibilities for the role of DPM-mediated inhibition in consolidation. One idea is that temporally-regulated inhibition of potentiated α′/β′ output could serve as a way of sharpening the transition of memory from one neuropil to another by suppressing activity in the brain area from which information has already been transferred. A second possibility is that a period of inhibition during consolidation is a way of preventing new, potentially interfering, information from being encoded in α′/β′ before the first memory is transferred to α/β. A third possibility is that the function of inhibition is actually to provide precisely timed rebound excitation to α′/β′. This would not have been seen in our imaging experiments due to the slow kinetics of ATP washout, but could actually result in feedback excitation of MB neurons.

All of these models imply that there is a very tight temporal ordering of activity within the circuit, with DPM neurons suppressing α′/β′ neuron activity in a narrow window either before or after their output function has been completed. It is important to note that none of these ideas is inconsistent with the demonstrated requirement for α′/β′ activity during the post-training consolidation period. GABA- and 5HT-mediated inhibition is not equivalent to the action of Shi[ts], which completely shuts off neurotransmission. Inhibition is often modulatory rather than switch-like and can even be compartment-specific—for example, it could serve to alter the ratio of activity in α′/β′ to that in α/β. More precise mapping of the connectivity and branch-specific activity in the MB neuropil will be required to develop a more detailed model.

## Coupling sleep and memory consolidation at the cellular level

Although investigators have speculated that memory consolidation and sleep interact, an actual understanding of how they are related at the circuit level has been elusive. This study is the first demonstration of a cellular- and circuit-level mechanism for the coupling of sleep and consolidation. The fact that memory and sleep are behaviorally linked even in the insect (*Ganguly-Fitzgerald et al., 2006*; *Donlea et al., 2011*) is evidence of the evolutionary importance of coupling these two processes. The simplicity of the cellular mechanism in *Drosophila*, using a single pair of neurons to carry out both functions, provides an example of how coupling can occur in a small nervous system and suggests a template for understanding it in larger brains.

# Materials and methods

## Fly stocks

Fly stocks were raised on modified Brent and Oster cornmeal-dextrose-yeast agar media (*Brent et al., 1974*) Per batch: 60 l $H_2O$, 600 g Agar, 1950 g flaked yeast, 1451 g cornmeal, 6300 g dextrose, 480 g NaKT, 60 g $CaCl_2$, 169 g Lexgard dissolved in ethanol. Flies were raised under a 12:12 light:dark cycle at 25°C except for animals carrying UAS-dTrpA1 which were raised at 22°C or flies carrying either *pTub-GAL80$^{ts}$* or *UAS-IVS-Syn21-Shi$^{ts}$* which were raised at 18°C. *UAS-P2X2* (*Lima and Miesenbock, 2005*), *lexAop-P2X2, lexAop-Epac1-camps (1A), lexAop-GCaMP3.0* (*Yao et al., 2012*), and *UAS-Epac1-camps(55A)* (*Shafer et al., 2008*), flies were kindly provided by Dr Orie Shafer. *UAS-Arclight* (*Cao et al., 2013*) was a gift from Dr Michael Nitabach, and the *5HT7-GAL4* flies (*Becnel et al., 2011*) were a gift from Dr Charles Nichols. The following lines have also been previously described: *20xUAS-IVS-GCaMP6M* (*Akerboom et al., 2012*), *UAS-Synapto-pHlorin* (SpH) (*Meisenbock et al., 1998*), *UAS-GCaMP3.0* (*Tian et al., 2009*), *UAS-dTrpA1* (chromosome 2 insertion site) (*Hamada et al., 2008*), *UAS-Rdl-RNAi 8-10J* (*Liu et al., 2007, 2009*), *20xUAS-IVS-Syn21-Shi$^{ts}$* (*Pfeiffer et al., 2012*), *UAS-mCD8-GFP* (*Lee and Luo, 1999*), *UAS-Brp-short-GFP* (*Schmid et al., 2008*; *Fouquet et al., 2009*), *UAS-Kir2.1* (*Baines et al., 2001*), *c316-GAL4* (*Waddell et al., 2000*), *c305a-GAL4* (*Krashes et al., 2007*), *NP2721-GAL4* (*Wu et al., 2011*), *VT64246-GAL4* (*Lee et al., 2011*), *MB247-GAL4* (*Zars, 2000*), *MB247-lexA* (*Pitman et al., 2011*), *R58E02-lexA* (*Liu et al., 2012*), *MB-GAL80* (*Thum et al., 2007*), *eyeless-GAL80* (*Chotard et al., 2005*), *ptub-GAL80$^{ts}$* (*McGuire et al., 2003*), *ptub>GAL80>* (*Gordon and Scott, 2009*), *LexAOP-Flp* (*Shang et al., 2008*), and *GH146-GAL4, NP5288-GAL4, NP2631-GAL4* (*Tanaka et al., 2008*). The following RNAi lines were obtained from the VDRC (*Dietzl et al., 2007*): *VGAT-RNAi* (X-stock #45917), *VGAT-RNAi* (II- stock #45916), *Trh-RNAi* (II-#105414), *GABA$_B$R3-RNAI* (III-#50176). The following RNAi lines have been functionally verified previously: *Rdl-RNAi* (*Liu et al., 2007*), *GABA$_B$R3-RNAi* (*Dahdal et al., 2010*).

The genetic intersectional method used to restrict expression to the MB is described in *Shang et al. (2008)*. MB restriction to the prime lobes with *c305a* is shown in *Perrat et al. (2013)*.

## SuperClomeleon flies

13xLexAOP-IVS-Syn21-SuperClomeleon expressing flies were generated using the SuperClomeleon construct designed by *Grimley et al. (2013)*. Gateway cloning (Invitrogen) was used to create the entry vectors pDonr221-13xLexAOP and pDonrP2rP3-Syn21-SuperClomelon-p10. A modified version of pBPGUw (*Pfeiffer et al., 2008*), pBPGUw-R1R3-p10 was used for gateway recombination and injection into flies where it was targeted to the attP40 landing site on the second chromosome. pBPGUw-R1R3-p10 was generated from the following modifications to pBPGUw: removal the DSCP (Drosophila Synthetic Core Promoter), replacement of the attR2 Gateway recombination site with attR3, and replacement of the weaker Hsp70 terminator with the stronger p10 terminator sequence (*Pfeiffer et al., 2012*).

The entry vector, pDonr221-13XLexAOP, was generated by PCR and Gateway cloning of the 13XLexAOP-Hsp70 TATA-IVS sequence from pJFRC19 (*Pfeiffer et al., 2010*), into the pDonr221 Gateway entry vector. The pDonrP2rP3-SuperClomeleon entry vector was generated by PCR and Gateway cloning of the SuperClomeleon sequence from pUC19-SuperClomeleon (*Grimley et al., 2013*) into the pDonrP2rP3 Gateway entry vector. Primers for the pDonr221 construct were designed by fusing Gateway attB1 and attB2 sequences upstream and downstream, respectively, of the 13xLex-AOP and IVS sequences.

Primers for pDonrP2rP3-SuperClomeleon were designed by fusing Gateway attB2r and attB3 sequences upstream and downstream, respectively, of the SuperClomeleon sequence. To enhance

expression, the 21bp Syn21 sequence (*Pfeiffer et al., 2012*) was added to the forward primer just upstream of SuperClomeleon. All PCRed and ligated sequences were verified by sequencing before injection into flies *Table 1*.

## Behavioral analysis

Individual virgin female flies were housed separately in 65 mm × 5 mm glass tubes (Trikinetics, Waltham, MA) containing 5% agarose with 2% sucrose. Parafilm with pinholes poked in it was used to cover the open end of each tube. 2- to 5-day old flies were entrained under standard light–dark conditions, with a 12 hr light phase and followed by 12 hr dark phase for 3–4 days prior to collection of data for sleep analysis. Locomotor activity was collected with DAM System monitors (Trikinetics) in 1 min bins as previously described (*Agosto et al., 2008*). Sleep was defined as bouts of uninterrupted inactivity lasting for five or more minutes (*Hendricks et al., 2000*; *Shaw et al., 2000*). Sleep/activity parameters (total sleep, mean sleep episode duration, maximum sleep episode duration, number of sleep episodes, and activity while awake) were analyzed for each 12-hr period of light or dark conditions and averaged across 3 days. Sleep analysis was conducted using an in-house Matlab program described previously (*Donelson et al., 2012*). Since we found that some, but not all sleep data were not normally distributed, we chose to use the less powerful, but more conservative Mann–Whitney/Wilcox ranked sum test rather than ANOVA and Tukey post-hoc tests, which assume data are normally distributed. As the Mann–Whitney/Wilcox ranked sum test is a pairwise test, this generates two p values, one for experimental vs the UAS control and one for the experimental vs GAL4 control line. In all figures only the most conservative/numerically greatest p value is reported.

For temperature-shift experiments, flies expressing either Shi$^{ts}$ or *Tub-GAL80$^{ts}$* were raised at 18°C, whereas flies expressing dTrpA1 were raised at 22°C. In all cases, baseline data were recorded at the respective rearing temperature (18°C or 22°C) and compared to sleep at the activation (dTrpA1) or suppression (Tub-GAL80$^{ts}$ and Shi$^{ts}$) temperature of 31°C. The effect of heat on sleep is highly sensitive to genotype. In order to assess heat-induced changes in sleep, first, the baseline sleep of each fly at either 18°C or 22°C was subtracted from sleep of the same fly at 31°C and this difference was averaged together across flies of the same genotype. Following baseline subtractions for each genotype, average GAL4 or UAS values were then subtracted from average experimental group values to obtain minutes of sleep gained or lost vs controls.

## Immunohistochemistry

For immunostaining, a standard fixation and staining protocol was used. Briefly, brains were dissected in ice-cold PBS and were fixed immediately after dissection for 15 min at room-temperature in 4% paraformaldehyde (vol/vol). Brains were incubated in PBS containing 0.5% Triton X-100, 10% NGS and primary or secondary antibodies for one night each with 3 × 15 min washes between each incubation. Brain samples were then mounted using Vectashield and were visualized by a Leica TCS SP5 confocal microscope with a 20×, 40×, or 63× objective lens. All images were taken sequentially to prevent bleed-through between channels. For colocalization, either mouse anti-GFP (1:200, Roche Applied Biosciences) or rabbit anti-GFP (1:1,000, Invitrogen A11122) was used together with transmitter-specific primary antibodies as follows: rabbit anti-dVGAT (1:400, (*Fei et al., 2010*), a kind gift from Dr D.E. Krantz), rabbit anti-GABA (1:200, Sigma; Cat. No. A2052), rabbit anti-Gad1 (1:500, kind gift from Dr FR Jackson), mouse anti-choline acetyltransferase (ChAT) (1:200, code 4B1 Developmental Studies Hybridoma Bank; [*Takagawa and Salvaterra, 1996*; *Yasuyama et al., 1996*]),

**Table 1.** Primer sequences

| Primers (5′-3′) | Sequence (Gateway sequences are in caps and vector sequences are lower case) |
|---|---|
| attB1-13xLexAOP forward | GGGGACAAGTTTGTACAAAAAAGCAGGCTATgcatgcctgcaggttactgtac |
| attB2-IVS reverse | GGGGACCACTTTGTACAAGAAAGCTGGGTAggccgcctgaagtaaaggataag |
| attB2r-Syn21-SuperClomeleon forward | GGGGACAGCTTTCTTGTACAAAGTGGAA*AACTTAAAAAAAAAAATCAAA*atggtgagcaagggcgagg, |
| attB3-SuperClomeleon reverse | GGGGACAACTTTGTATAATAAAGTTGCttaaagcttcttgtacagctcgtccatg |

rabbit anti-serotonin (1:1,000, Sigma S5545), and mouse anti-tyrosine hydroxylase (TH) (1:500, Immunostar 22,941). Alexa Fluor 488 and 635 anti-mouse or anti-rabbit secondary antibodies (1:200, Invitrogen) were used to visualize staining patterns. Alexa 488 was always used to label GFP so that any residual endogenous GFP fluorescence would be of a similar wavelength as the dye and would not bleed through to the 633 wavelength channel.

## Image processing and intensity correlation analysis (ICA)

All image processing was done using the freely available FIJI (IMAGEJ) software and plugins (*Schindelin et al., 2012*). Background was subtracted from all confocal stacks prior to further processing. All images are sums or maximum intensity Z-projections of the relevant confocal slices. Quantification of cells with positive/negative staining was done by visually comparing colocalization of GFP and antibody staining in multiple individual Z-slices. Cell bodies/MB lobe sets where high background prevented interpretation of staining were excluded. For the images presented in *Figure 3*, Intensity Correlation Analysis (ICA) was also performed to assess spatial colocalization of staining between channels (*Li et al., 2004*). Rather than only comparing visible overlap of the absolute fluorescence intensity in each channel (red plus green equals yellow), which is subject to viewer bias and differences in staining intensity between channels, this method determines whether changes in staining intensity covary or are correlated between channels. This provides an objective and spatially specific representation of colocalization. ICA analysis generates a correlation/colocalization value for each pixel defined by the Product of the Differences from the Mean (PDM) that is, PDM= (red intensity-mean red intensity) × (green intensity – mean green intensity). PDM values for each pixel can then be visualized as an image showing positive intensity correlation (PDM>0) and negative intensity correlation (PDM<0). Relative PDM value scales are shown on each figure generated from ICA analysis.

## Functional fluorescence imaging

Adult hemolymph-like saline (AHL) consisting of (in mM) 108 NaCl, 5 KCl, 2 CaCl$_2$, 8.2 MgCl$_2$, 4 NaHCO$_3$, 1 NaH$_2$PO$_4$-H$_2$O, 5 trehalose, 10 sucrose, 5 HEPES; pH 7.5 (*Wang, 2003*) was used to bathe the brain, as previously described (*Wang, 2003*; *Shang et al., 2011*). For SuperClomeleon experiments, the AHL pH was increased to 7.7 to optimize response magnitude. Test compounds adenosine 5′-triphosphate magnesium salt (ATP), carbamoylcholine chloride (CCh), picrotoxin (PTX), and γ-aminobutyric acid (GABA) were purchased from Sigma–Aldrich (St Louis, MO), serotonin hydrochloride (5HT) was purchased from Tocris Bioscience (Minneapolis, MN), and tetrodotoxin (TTX) was purchased from Abcam Biochemicals (Cambridge, England). ATP, CCh, and TTX were dissolved in milliQ water and frozen as aliquot stocks, which were then prepared for experiments by dilution in AHL. PTX was dissolved and frozen in DMSO aliquots, which were diluted in AHL for experiments. All solutions used in experiments with PTX were prepared with the same percentage of DMSO. GABA and 5HT were dissolved directly in AHL immediately prior to the experiment, and 5HT was kept in light-shielded containers to prevent degradation.

Imaging experiments were performed using a naked brain preparation. Flies were anesthetized on ice, and brains were dissected into cool AHL. Dissected brains were then pinned to a layer of Sylgard (Dow Corning, Midland, MI) silicone under a small bath of AHL contained within a recording/perfusion chamber (Warner Instruments, Hamden, CT). Brains expressing GCaMP3.0, Arclight, and SpH were allowed to settle for 5 min after dissection to reduce movement. These brains were then exposed to fluorescent light for approximately 30 s before imaging to allow for baseline fluorescence stabilization, while brains expressing the FRET sensors Epac1-camps or SuperClomeleon were exposed an extra 5 min to minimize differences in photobleaching rates between the CFP and YFP fluorophores, as YFP has been described to photobleach more slowly than CFP (*Shafer et al., 2008*; *Pirez et al., 2013*). Perfusion flow was established over the brain with a gravity-fed ValveLink perfusion system (Automate Scientific, Berkeley, CA). ATP, CCh, GABA, or 5HT were delivered by switching perfusion flow from the main AHL line to another channel containing diluted compound after 30 s of baseline recording for desired durations followed by a return to AHL flow. To control for the effects of switching channels, a vehicle control trial was performed by switching to another line containing AHL for the same duration as the experimental trial.

Imaging was performed using an Olympus BX51WI fluorescence microscope (Olympus, Center Valley, PA) under an Olympus x40 (0.80W, LUMPlanFl) or x60 (0.90W, LUMPlanFl) water-immersion objective, and all recordings were captured using a charge-coupled device camera (Hamamatsu ORCA

C472-80-12AG). For GCaMP3.0, Arclight, and SpH imaging, we used the following filter set (Chroma Technology, Bellows Falls, VT): excitation, HQ470/x40; dichroic, Q495LP; emission, HQ525/50m. For EPAC and SuperClomeleon, a 86002v1 JP4 (436; Chroma Technology) excitation filter was used, and emitted light from the CFP and YFP flurophores was separated using a splitter (Photometrics DV$^2$ column) with the emissions filters D480/30m and D535/40m (Photometrics, Tucson, AZ), which allowed for simultaneous collection from both fluorescence channels. Frames were captured at 2 Hz with 4× binning for either 2 min or 4 min using μManager acquisition software (*Edelstein et al., 2010*). Neutral density filters (Chroma Technology) were used for all experiments to reduce light intensity to limit photobleaching.

Although there are tools for temporally controlled activation such as dTrpA1 or Channelrhodopsin (ChR2) that are well-characterized, we utilized P2X2 receptors for the majority of our experiments. Activation wavelengths for ChR2 overlap with those of many fluorescent sensors such as GCaMP and EPAC, so ChR2 could not be used in this circuit due to the close proximity of the cells and processes. Applying heat to activate dTrpA1 causes changes in refractive index, which disrupt focus, necessitating manual focus correction in the cases where we used this technique.

Regions of interest (ROIs) were selected within the horizontal or vertical lobes of the MBs or, in the case of DPM neurons, over the horizontal projections to the MBs. Figures depict responses in horizontal lobes or projections; however, similar results were observed in the vertical lobes when noted in the figure captions. For recordings using GCaMP3.0, Arclight, and SpH, ROIs were analyzed using custom software developed in ImageJ (*Schindelin et al., 2012* and National Institute of Health, Bethesda, MD). Briefly, the percent change in fluorescence over time was calculated using the following formula: $\Delta F/F = (F_n - F_0)/F_0 \times 100\%$, where $F_n$ is the fluorescence at time point n, and $F_0$ is the fluorescence at time 0. For GCaMP3.0 and SpH, maximum fluorescence change values were determined as the maximum percentage change observed for each trace over the entire duration of each imaging experiment. For Arclight, because increases in voltage are represented as decreases in fluorescence, maximum fluorescence change values were determined as the minimum percentage change. Maximum values for each group were then averaged to calculate the mean maximum change from baseline.

For recordings using EPAC or SuperClomeleon, ROIs were analyzed using custom software developed in MATLAB (The MathWorks, Natick, MA). This analysis package is provided in *Source Code 1*. Briefly, identical ROIs were selected from both the CFP and YFP emissions channels, and the fluorescence resonance energy transfer (FRET) signal (YFP/CFP ratio) was calculated for each time point and normalized to the ratio of the first time point. The relative cAMP changes were determined by plotting the normalized CFP/YFP ratio (percentage) over time. As with GCaMP3.0 and SpH, the average maximum percent change values were determined as the mean maximum values for each group.

Statistical analyses were performed using MATLAB (The MathWorks). A Kruskal–Wallis one-way ANOVA was used to determine statistical significance between the experimental group and the two negative controls. In cases in which there was significance in the ANOVA, a Mann–Whitney U test (also known as Wilcoxon rank-sum test) was used to determine the significance between the experimental group and each negative control. In all figures only the most conservative/numerically greatest p value is reported. Results are expressed as means $\pm$ standard error of the mean (SEM).

## Acknowledgements

This work was supported by NIH grants R01 MH067284 (to LCG) and F31 NS086764 (to BLC). The Brandeis imaging facility was supported by NIH grant P30 NS045713. We would like to thank Michael Nitabach and Orie Shafer for sharing lines before publication and Thomas Christmann for help with figures.

## Additional information

### Competing interests

LCG: Reviewing editor, *eLife*. The other authors declare that no competing interests exist.

### Funding

| Funder | Grant reference number | Author |
|---|---|---|
| National Institute of Mental Health | R01 MH067284 | Leslie C Griffith |

| Funder | Grant reference number | Author |
| --- | --- | --- |
| National Institute of Neurological Disorders and Stroke | F32 NS045713 | Bethany L Christmann |

The funders had no role in study design, data collection and interpretation, or the decision to submit the work for publication.

## Author contributions

PRH, BLC, Conception and design, Acquisition of data, Analysis and interpretation of data, Drafting or revising the article; LCG, Conception and design, Analysis and interpretation of data, Drafting or revising the article

## Additional files

### Supplementary file

• Source code 1. Custom software developed in MATLAB.

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
