## [Decision Letter]

Thank you for sending your work entitled “A single anatomical locus links sleep to memory consolidation in *Drosophila melanogaster*” for consideration at *eLife*. Your article has been favorably evaluated by K. VijayRaghavan (Senior editor), Graeme Davis (Reviewing editor) and 3 reviewers, one of whom, Amita Sehgal, has agreed to reveal her identity.

Three reviewers have read and discussed your manuscript in detail. All are well-established investigators with expertise in this area of research. All agree that this is an interesting and important area of research and that your manuscript has the potential to make a significant advance that would appeal to the broad audience of *eLife*. In general, there is agreement that the data are nicely presented and the manuscript is well written. There are a few major concerns that the reviewers have identified that require attention as well as several minor concerns that you can address at your discretion.

Major concerns:

1) The title of the manuscript emphasizes a single anatomical locus, but this is not clearly demonstrated. Perhaps the authors can restate the major finding in the title in some other way. A more accurate concept to emphasize is that the authors have identified an anatomical substrate for the control of both processes.

2) The data in Figure 1 need to be revisited. This is critical to the success of the manuscript. Silencing of DPM neurons in Figure 1 produces a very weak effect, which could reasonably be the result of manipulating a small number of neurons. However, there is a concern that the reported significant change is due to two issues: A) an atypical method used to calculate sleep loss and B) the comparison that is being made. Specifically, it is difficult to accept the claim that sleep is reduced when the experimental curves completely overlap with one or both controls at 31 degrees (except for the tiny effect on the second night of silencing). The experimental genotype is more affected by the temperature change, but only if you accept that sleep in the GAL4 control is lower than the experimental at baseline. Given that sleep in the two genotypes is the same following recovery, the lower sleep in the control is not a consistent phenotype. If anything, following recovery the control should be even lower than the experimental because the experimental has lost sleep and may be undergoing rebound. The fact that they're the same means there's something strange going on with that control. Ideally, the control and experiments would have the same sleep at baseline, and then one could compare sleep between genotypes at the higher temperature. One possibility to help address this issue would be for the authors to include another shift to 31 degrees following a night of equal sleep between the controls and experimental animals at 18.

3) The staining in Figure 2 looks very nice, but as many of these antibodies can be problematic, the authors should show controls for the antibodies used (positive control for ChaT and negative for GABA and GAD). See also comment 6 below.

4) The absence of direct physiological evidence for an inhibitory connection between the DPM and the α' and β' neurons is unfortunate, but it seems clear that this is a shortcoming of the available tools for measuring inhibition deep in the *Drosophila* brain rather than a technical shortcoming of this study. In the Results section, describing the negative physiological results, the authors state: “Unfortunately, these fluorescent sensors are often not effective at reporting inhibition unless there is some basal level of activity in the circuit.” The authors should cite instances where this has been shown here in the Results section or direct the reader to the citations present in the Discussion.

5) It would be nice to see a more detailed description of where the various GAL4 lines are expressed outside the DMP neurons in Figure 1—figure supplement 1. It would also be useful for the non-specialist reader if the DPM neurons were indicated in each of the micrographs in this supplementary figure.

6) The data regarding synaptic localization of VGAT and Bruchpilot is a concern. These are broadly overlapping, relatively weak, immunostaining profiles in complex neuropil. Related to control for antibody staining, could the authors knock down VGAT and provide a control for this overlap?

7) The decisions to include data in the primary text versus the supplementary data seem a bit arbitrary in some cases. As a consequence, the reader can be lead to speculate why these decisions were made. For example, the authors dismiss the APL > *vGAT-RNAi* data as having a weak effect (Figure 5–figure supplement 1), but this effect seems at least as strong as what is seen with their other manipulations that they consider significant. I don't think that their argument—that because they also see an increase in activity in this experiment means the effect on sleep is irrelevant—is a valid one. Sleep could be affected in a meaningful way even if locomotor activity is also affected. As another example, the RNAi mediated knockdown of 5HT synthesis and GABA vesicle loading provide very strong evidence for this manuscript's conclusion that the DPM neurons inhibit the MBs. It would be appropriate for the authors to show the data for the milder behavioral effects of *TRH-RNAi* knockdown using *NP2721-GAL4*, since, based on Figure 1—figure supplement 1, this line is significantly more specific to the DPM neurons.

[Editors' note: further revisions were requested prior to acceptance, as described below.]

Thank you for resubmitting your work entitled “A single pair of neurons links sleep to memory consolidation in *Drosophila melanogaster*” for further consideration at *eLife*. Your revised article has been favorably evaluated by K. VijayRaghavan (Senior editor), a Reviewing editor, and 2 of the original reviewers. The manuscript has been improved but there are some remaining issues that need to be addressed before acceptance, as outlined below:

All of the reviewers and the Reviewing editor agree that the authors have responded to the reviewers’ criticism effectively with considerable inclusion of new data and thoughtful revision. However, one major point regarding the data in Figure 1 remains an unresolved issue that the reviewers feel could be easily addressed and would largely resolve any remaining concerns. After addressing this remaining issue, the reviewers agree that the manuscript is acceptable for publication at *eLife* without necessitating further review. The reviewers and Reviewing editor have discussed this remaining issue extensively and suggest the following solution:

The new data that were provided in the response to the reviewer criticism regarding Figure 1 seem to show that there is no baseline sleep defect on day/night 2, the first experimental night. The effect on day/night 2 should be quantified and shown in the main figure in the body of the manuscript. Further, the data for day/night 4 and 5 suggest a very small effect. All these data need to be shown in the main body of Figure 1 (instead of the current version of Figure 1), and the magnitude of this effect should be noted and discussed in the text. Further, it is very important that the statistical significance of the proposed effect in these new data can be evaluated and reported.

---

## [Author Response]

*1) The title of the manuscript emphasizes a single anatomical locus, but this is not clearly demonstrated. Perhaps the authors can restate the major finding in the title in some other way. A more accurate concept to emphasize is that the authors have identified an anatomical substrate for the control of both processes*.

We have changed the title to the more descriptive “A single pair of neurons links sleep to memory consolidation in *Drosophila melanogaster*”. We hope this addresses the concerns of the reviewers.

*2) The data in*
Figure 1
*need to be revisited. This is critical to the success of the manuscript. Silencing of DPM neurons in*
Figure 1
*produces a very weak effect, which could reasonably be the result of manipulating a small number of neurons. However, there is a concern that the reported significant change is due to two issues: A) an atypical method used to calculate sleep loss and B) the comparison that is being made. Specifically, it is difficult to accept the claim that sleep is reduced when the experimental curves completely overlap with one or both controls at 31 degrees (except for the tiny effect on the second night of silencing). The experimental genotype is more affected by the temperature change, but only if you accept that sleep in the GAL4 control is lower than the experimental at baseline. Given that sleep in the two genotypes is the same following recovery, the lower sleep in the control is not a consistent phenotype. If anything, following recovery the control should be even lower than the experimental because the experimental has lost sleep and may be undergoing rebound. The fact that they're the same means there's something strange going on with that control. Ideally, the control and experiments would have the same sleep at baseline, and then one could compare sleep between genotypes at the higher temperature. One possibility to help address this issue would be for the authors to include another shift to 31 degrees following a night of equal sleep between the controls and experimental animals at 18*.

The first issue is our analysis method. We actually developed this because we were dissatisfied with the commonly used “% change” method used in previous papers from our lab since when the baseline is small, really minor changes in sleep can be exaggerated. Our method avoids this pitfall, and we believe it is a more accurate representation of the changes that occur after a temperature shift. This method is not all that unusual. In fact, our analysis method is identical to what has been published by others (including [44]) except that in our case, sleep changes are expressed as minutes of sleep gained or lost vs. controls rather than as a percent change from controls. The final step of dividing the sleep of the experimental group by that of each control group used by other groups does not change the control or experimental data distributions or relative differences between them since both the experimental and control groups are divided by the control group values. It is simply an additional scaling step that converts minutes of sleep lost or gained into a net percentage. We feel that our analysis method is in line with the field but is a more accurate report of changes in sleep.

The second issue is the data set shown in Figure 1. Firstly we completely agree that the *NP2721-GAL4/+* control in this experiment is not normal. It appears to be changed in a lasting way by the 31^o^ C heat treatment so that it does not return to its relative pre-heat baseline. Unfortunately, we have outcrossed this line 6 generations to *Canton S* without much change in this baseline behavior.

To demonstrate that this is not affecting our results, we have carried out the double heat treatment experiment suggested by the reviewers, and it is shown below (Figure 8). As you can see, the GAL4 control once again gains sleep after a day of heating and becomes almost the same as the experimental line and the UAS control in the first recovery day (Day3). Importantly however, if you apply a second heat treatment, you still get a relative sleep loss in the experimental line if you now use that first recovery day (Day 3) as the baseline for the second heating period (Day 4/5). We find that this particular double heating treatment also appears to result in daytime sleep gains, but this is likely due to the sleep loss occurring during the daytime on the day used as a baseline (Day 3). This sleep loss likely reflects an ongoing suppression of the DPMs (note it is also seen in new Figure 1). This is unusual and interesting, but its investigation is beyond the scope of this paper.Author response image 1.

While this does not really resolve the nature of this behavior of this control strain, we feel that it does demonstrate that it is not relevant to the DPM phenotype since dTrpA1-dependent night-time sleep loss is seen no matter what the GAL4 control baseline is doing. The overall case for DPMs having a role in production of baseline night-time sleep is very strong and does not rest solely on any one experiment. The data in Figure 1—figure supplement 2 show that acute induction of Kir by a temperature shift produces a clean reduction in sleep compared to both GAL4 and UAS controls. This experiment was done with a third DPM-*GAL4* (*VT64246*), a line that does not have the same baseline sleep phenotype as *NP2721*. This demonstrates that with two distinct GAL4s and two distinct activity-reducing manipulations, we can see a qualitatively similar reduction in sleep when we inhibit DPM output.

*3) The staining in*
Figure 2
*looks very nice, but as many of these antibodies can be problematic, the authors should show controls for the antibodies used (positive control for ChaT and negative for GABA and GAD). See also comment 6 below*.

We have updated the image shown in Figure 3 to show that our staining, although it does not detect significant ChAT signals in the DPMS, is able to detect ChAT in a large population of neurons in the brain, consistent with published reports of this Ab being specific for cholinergic neurons ([87] and [107]). The negative control for the anti-GABA antibody we used is shown in [61] where they demonstrate that expression of GAD-RNAi reduced anti-GABA staining in a cell-specific manner. The anti-GAD antibody we used was verified by showing that there was no staining of homozygous null embryos (Featherstone et al., 2000).

*4) The absence of direct physiological evidence for an inhibitory connection between the DPM and the α' and β' neurons is unfortunate, but it seems clear that this is a shortcoming of the available tools for measuring inhibition deep in the* Drosophila *brain rather than a technical shortcoming of this study. In the Results section, describing the negative physiological results, the authors state: “Unfortunately, these fluorescent sensors are often not effective at reporting inhibition unless there is some basal level of activity in the circuit.” The authors should cite instances where this has been shown here in the Results section or direct the reader to the citations present in the Discussion*.

Since the initial submission of this paper we have been able to develop a method to visualize changes in intracellular chloride in *Drosophila* using SuperClomeleon (37). The new Figure 5 shows that both bath application of GABA and activation of DPMs result in an increase in intracellular chloride which are decreased by concurrent application of picrotoxin. We have modified the text and the discussion of other sensors accordingly. This result substantially strengthens the paper.

As for citations showing no effect of inhibition with calcium and cAMP sensors, we were not able to come up with any. What we can (and do) cite is that all the studies in which inhibition was successfully documented used some stimulus to activate the system before the “inhibitory” manipulation. We have therefore modified our wording in that section to indicate that it is “likely” that there has to be some basal activity. Because we do not see an inhibitory response using calcium and cAMP, despite our finding that DPM activation causes MB chloride influx, we believe that this itself provides decent evidence that these reporters can fail to show a negative response in the context of documented inhibition.

*5) It would be nice to see a more detailed description of where the various GAL4 lines are expressed outside the DMP neurons in*
Figure 1—figure supplement 1*. It would also be useful for the non-specialist reader if the DPM neurons were indicated in each of the micrographs in this supplementary figure*.

Figure 1—figure supplement 1 has been annotated to label major regions of ectopic expression, and an asterisk has been added to denote the DPM neurons in each panel.

*6) The data regarding synaptic localization of VGAT and Bruchpilot is a concern. These are broadly overlapping, relatively weak, immunostaining profiles in complex neuropil*. *Related to control for antibody staining, could the authors knock down VGAT and provide a control for this overlap?*

The reviewers are correct that believable colocalization can be difficult when the antigens are broadly expressed in a complex neuropil. This is why we chose to use a correlation algorithm as opposed to looking at overlap as most studies do. Rather than comparing visible overlap of the absolute fluorescence intensity in each channel (e.g. red plus green equals yellow), which is subject to viewer bias and differences in staining intensity between channels, this method determines whether changes in staining intensity co-vary or are correlated between channels.

This provides an objective and spatially specific representation of colocalization. ICA analysis generates a correlation/colocalization value for each pixel defined by the Product of the Differences from the Mean (PDM). PDM= (red intensity - mean red intensity)×(green intensity – mean green intensity). PDM values for each pixel can then be visualized as an image showing positive intensity correlation (PDM>0) and negative intensity correlation (PDM<0). Relative PDM value scales are shown on each figure generated with ICA analysis.

This method produces an image with a higher confidence of colocalization than standard overlap methods. The ability of this method to discriminate true colocalization on a diffuse background is highlighted by the data shown in Figure 3. In the presence of diffuse local ChAT staining, we see no evidence of colocalization to DPMs. This is the most rigorous control that you could ask for in terms of validation of the technique. This, and the consistency of the VGAT data set with the validated GABA and GAD antibodies, gives us confidence that this VGAT/Brp colocalization result is correct and we feel comfortable leaving it in the paper.

*7) The decisions to include data in the primary text versus the supplementary data seem a bit arbitrary in some cases. As a consequence, the reader can be lead to speculate why these decisions were made. For example, the authors dismiss the APL >* vGAT-RNAi *data as having a weak effect (Figure 5–figure supplement 1), but this effect seems at least as strong as what is seen with their other manipulations that they consider significant. I don't think that their argument—that because they also see an increase in activity in this experiment means the effect on sleep is irrelevant—is a valid one. Sleep could be affected in a meaningful way even if locomotor activity is also affected. As another example, the RNAi mediated knockdown of 5HT synthesis and GABA vesicle loading provide very strong evidence for this manuscript's conclusion that the DPM neurons inhibit the MBs. It would be appropriate for the authors to show the data for the milder behavioral effects of* TRH-RNAi *knockdown using* NP2721-GAL4*, since, based on*
Figure 1—figure supplement 1*, this line is significantly more specific to the DPM neurons.*

The first question here is the role of APLs. We did not mean to imply that the locomotor phenotype made APL sleep effects irrelevant. What we were trying to convey is that the sleep effects of down regulating GABA in these cells is qualitatively different from what happens when you decrease GABA in DPMs, indicating that the two cells are functioning independently. We have tried to make this clearer in the revised text.

We have also provided additional data in Figure 1 (new panel C) that more definitively exclude APL as a cause of sleep effects after activation of DPMs. We show in the new data that coexpression of Shi^ts^ with dTrpA1 in DPMs completely blocks neuronal activation-dependent sleep increases. Because the effects of Shi^ts^ are limited to DPM, this strongly suggests that APL sleep phenotypes are separate from those of DPMs and not relevant to the DPM story. We have amended the text to make this distinction clearer.

The second issue was relegation of part of the 5HT data to supplement. We have taken the reviewers’ suggestion and moved it into Figure 6.

[Editors' note: further revisions were requested prior to acceptance, as described below.]

*All of the reviewers and the Reviewing editor agree that the authors have responded to the reviewers’ criticism effectively with considerable inclusion of new data and thoughtful revision. However, one major point regarding the data in*
Figure 1
*remains an unresolved issue that the reviewers feel could be easily addressed and would largely resolve any remaining concerns. After addressing this remaining issue the reviewers agree that the manuscript is acceptable for publication at* eLife *without necessitating further review*. *The reviewers and Reviewing editor have discussed this remaining issue extensively and suggest the following solution:*

*The new data that were provided in the response to the reviewer criticism regarding*
Figure 1
*seem to show that there is no baseline sleep defect on day/night 2, the first experimental night. The effect on day/night 2 should be quantified and shown in the main figure in the body of the manuscript. Further, the data for day/night 4 and 5 suggest a very small effect. All these data need to be shown in the main body of*
Figure 1
*(instead of the current version of*
Figure 1*), and the magnitude of this effect should be noted and discussed in the text. Further, it is very important that the statistical significance of the proposed effect in these new data can be evaluated and reported*.

We have included the control “two cycle” experiment in Figure 1, with statistical analysis as requested, and a commentary in the Results section. It turns out that both the first and second temperature shifts caused small but statistically significant sleep loss, indicating that the aberrant behaviour of the GAL4 control’s baseline was immaterial to the original finding.

We have also added a supplementary figure (new Figure 1—figure supplement 3) that shows the same result as old Figure 1 with an independent DPM GAL4 driving Shi^ts^, further supporting the finding that inactivation of DPMs causes small losses in night-time sleep.

In putting this revision together, we became very concerned that the substitution of the experiment with multiple cycles for the original data was going to be very confusing to the reader since there was no a priori reason for us to have done the experiment with that kind of protocol. In the end, we decided that, for the sake of clarity, we would deviate from your request in one respect: we left the original Figure 1 data in as Figure 11 with the control experiment as Figure 1. This allowed us to explain in text the rationale for the two heat cycles and make a stronger point that while the night-time sleep changes with DPM inactivation are very small, they are independent of pre-inactivation baseline (first vs. second heat cycle with *NP2721*), GAL4 driver (*NP2721-GAL4* in original Figure 1 and new c316 data), and activity manipulation (Shi^ts^ and Kir, which was in the previous version of the paper as a Figure 1—figure supplement 1).

I think that, given the multiple approaches we have used to flesh out this relatively minor point, I would still prefer to leave Figure 1 as it was (or substitute in the c316-GAL4 data in the new Figure 1—figure supplement 3) and put the cycling experiment into a supplementary figure. I believe it would be clearer for the reader. But given the insistence of the reviewers, the new version conforms (mostly!) to the instructions.